# Status and Future Scope of Soft Nanoparticles-Based Hydrogel in Wound Healing

**DOI:** 10.3390/pharmaceutics15030874

**Published:** 2023-03-08

**Authors:** Marcel Henrique Marcondes Sari, Alexandre de Fátima Cobre, Roberto Pontarolo, Luana Mota Ferreira

**Affiliations:** 1Postgraduate Program in Pharmaceutical Sciences, Federal University of Santa Maria, Santa Maria 97105-900, Brazil; 2Postgraduate Program in Pharmaceutical Sciences, Federal University of Paraná, Curitiba 80210-170, Brazil; 3Pharmacy Department, Federal University of Paraná, Curitiba 80210-170, Brazil

**Keywords:** gel, cutaneous lesions, nanostructures, polymers, natural products

## Abstract

Wounds are alterations in skin integrity resulting from any type of trauma. The healing process is complex, involving inflammation and reactive oxygen species formation. Therapeutic approaches for the wound healing process are diverse, associating dressings and topical pharmacological agents with antiseptics, anti-inflammatory, and antibacterial actions. Effective treatment must maintain occlusion and moisture in the wound site, suitable capacity for the absorption of exudates, gas exchange, and the release of bioactives, thus stimulating healing. However, conventional treatments have some limitations regarding the technological properties of formulations, such as sensory characteristics, ease of application, residence time, and low active penetration in the skin. Particularly, the available treatments may have low efficacy, unsatisfactory hemostatic performance, prolonged duration, and adverse effects. In this sense, there is significant growth in research focusing on improving the treatment of wounds. Thus, soft nanoparticles-based hydrogels emerge as promising alternatives to accelerate the healing process due to their improved rheological characteristics, increased occlusion and bioadhesiveness, greater skin permeation, controlled drug release, and a more pleasant sensory aspect in comparison to conventional forms. Soft nanoparticles are based on organic material from a natural or synthetic source and include liposomes, micelles, nanoemulsions, and polymeric nanoparticles. This scoping review describes and discusses the main advantages of soft nanoparticle-based hydrogels in the wound healing process. Herein, a state-of-the-art is presented by addressing general aspects of the healing process, current status and limitations of non-encapsulated drug-based hydrogels, and hydrogels formed by different polymers containing soft nanostructures for wound healing. Collectively, the presence of soft nanoparticles improved the performance of natural and synthetic bioactive compounds in hydrogels employed for wound healing, demonstrating the scientific advances obtained so far.

## 1. Introduction

Skin is an interface between the external and internal body environments and is structurally formed by three layers: epidermis, dermis, and hypodermis. The cutaneous tissue is responsible for preventing water and nutrient loss, representing approximately 15% of body weight [1]. Remarkably, skin plays an essential role in protecting the human body against harmful agents, such as toxic substances, ultraviolet radiation, mechanical and physical damage, and pathogenic microorganisms, acting as a physical-mechanical and immunological barrier [2]. Thus, changes in skin integrity caused by any trauma are called wounds, which can be differently named according to various injury factors, for instance, bruises, incisions, and cuts [3,4]. The lesions are generally induced by external factors, such as surgery, mechanic impact, heat, cold, chemicals, and biological factors, including blood supply disorders, diabetes, and leishmaniosis [5,6,7]. Hindering the healing process, infections are hazardous to tissue, and delay tissue restoration, increasing expenses, treatment duration, and the risk of complications [8].

The selection of a wound dressing product must consider the lesion stage, the application frequency, the cost of care, and the need for any pharmacological intervention (such as antibiotics and painkillers) [9,10]. Adequate wound dressings should be non-toxic, non-sensitizing, and present suitable mechanical and rheological properties to promote easy application at the desired site [11]. Additionally, avoiding infection and maintaining wound humidity is essential to promote healing [12]. Currently, various dressings are available for managing acute and chronic wounds, where hydrogels are particularly highlighted given their potential advantages [13].

Hydrogels are composed of three-dimensional hydrophilic network structures that quickly swell when interacting with water, assuming a semisolid form [14]. Generally, the amount of water exceeds 90% in the hydrogel matrix, providing adequate conditions to maintain wound moisture [15]. Then, due to the water content, hydrogel dressings can simplify the healing stages given their suitable bioadhesion, facilitating the dressing removal without secondary injury, reducing the discomfort and the infection risk. Some hydrogel dressings are designed to be transparent, allowing clinical monitoring of the healing stages without removing the dressing [16]. Furthermore, the hydrogels can be combined with drugs and accelerate the healing process by associating both the advantages of healing devices and the pharmacological properties of active substances [15]. 

Despite the potentialities, some features of the hydrogels require optimization, such as mechanical properties, biological activity, and pharmacokinetic parameters of the drug incorporated in the matrix [13]. Therefore, a great deal of effort has been invested in prospecting novel approaches for overcoming these restrictions, which includes the development of hydrogels containing nanocarriers [17,18,19]. The nanoparticles can be classified according to their composition and structural organization, including hard nanoparticles, which are made of inorganic materials, and nanocarriers composed of organic materials, named soft nanoparticles [20]. Concerning wound healing purposes, soft nanoparticles are more appropriate than hard particles because of their biocompatibility, biodegradability, and physicochemical properties that are susceptible to size and shape adaptation when facing different biological conditions such as pH, ionic strength, and pressure [21].

In this sense, the development of formulations exploring nano-based systems for loading drugs has increased considerably in previous years [22,23]. Numerous studies have shown that the association of nanotechnology and wound dressing devices is a promising strategy, especially when nanostructured systems are used to improve the performance of active compounds throughout the skin [24]. Furthermore, nano-based formulations present improved bioadhesion, physicochemical stability, and biocompatibility [25], and have sparked scientific and pharmaceutical interest in recent years. 

Given the above, this scoping review seeks to shed light on the state-of-the-art regarding the association of soft nanoparticles, mainly composed of lipids and biocompatible and biodegradable polymers, and hydrogels designed for wound healing. The review approaches general aspects of the healing process, current status and limitations of hydrogels containing non-encapsulated drugs, and soft nanostructures-based hydrogels designed for wound healing. Therefore, this review hopes to contribute to improving the pharmacological management of wounds, demonstrating the scientific advances obtained so far and stimulating researchers to identify future avenues worth investigating.

## 2. Wound Healing Process

Wounds are considered a serious worldwide problem, presenting significant rates of morbidity and mortality, and increasing the costs of treating health complications [26,27]. Technically, a wound is any anatomical/cellular discontinuity of tissue that impairs skin functions, usually produced by external traumatic action [3]. Wound healing is a complex physiological process consisting of four overlapping phases where distinct molecular and cellular pathways are involved [28].

### 2.1. Wound Types

Intact skin tissue has important functions such as protection against microorganisms and toxins, water loss prevention, and thermoregulation [3]. A trauma may induce a loss of skin integrity that impairs its protective function, allowing the entry of microorganisms causing systemic or local inflammation and infection [29]. In this sense, wounds can be classified in several ways, according to etiology, morphology, contamination degree, healing evolution stage, and characteristics of bed and exudate, among others [30]. Acute injuries, such as surgical incisions, usually heal within days or weeks, and their extremities move closer together, decreasing infection rates [28]. Unlike acute lesions, chronic lesions last more than three months and do not evolve along the normal sequence of repair. The lesions’ extremities do not approach each other, increasing the infection risk and delaying healing time, including deep ulcers of pressure, peripheral vascular, and venous or arterial ulcers [10].

Accidental traumatic wounds can be caused by avulsions, lacerations, burns, or punctures, including bites, firearms, and piercing objects, which are susceptible to infection and demand long-term care [31]. Three important criteria are considered for wound classification: the wound type (open or closed), the degree, and the duration of contamination [30]. Wounds are classified as closed when the superficial layer of skin remains intact, resulting in wound protection against any contamination, but underlying tissues may be injured. Bruises and contusions are examples of these wounds. In contusions/fractures the underlying skin may appear intact but later becomes non-viable, resulting in an open wound [32]. Conversely, open wounds represent an interruption of the skin level or mucous membranes, where the following types are highlighted [33]: (a) avulsion—brutal separation of tissues and their annexes occurs, and (b) incisions. Another important type is caused by burns, which are secondary injuries from accidents involving excess heat or cold, chemicals, or electricity that can damage the skin and adjacent tissues [34].

There are some lesions caused by infective diseases, such as Leishmaniasis and its cutaneous form, frequently observed in South America [35]. Cutaneous Leishmaniosis is one of the most relevant neglected infections that produces skin damage followed by a persistent scar [35]. The disease can be caused by different Leishmania species, including *L. tropica*, *L. aethiopica*, *L. major*, and *L. braziliensis* [35]. Generally, the cutaneous lesions appear on the exposed parts of the body, such as the face, arms, and legs, several weeks or months after exposure to the parasite. The lesions progress from papules to nodular plaques that may evolve into ulcerative lesions, which are painless, but can become painful after infections. The ulcerative lesion heals spontaneously, resulting in atrophic scarring formation [36]. Leishmaniasis treatment is complex because of the narrow therapeutic window of anti-leishmaniasis drugs and parasite resistance [36].

Lastly, chronic wounds are defined as any interruption in the continuity of body tissue that hinders the healing process and presents long-last permanence [10]. Considered public health problems, these lesions affect 5% of the adult population in the western world, having high rates of recurrence and representing critical costs for health services [37]. Several local and systemic factors can hinder tissue healing, such as age, stress, medications, nutrition, and infection [10]. Among these factors, diabetes is one of the most important and prevalent causes of wound development given that the disease is characterized by an impairment in the general healing process, facilitating the occurrence of diabetic ulcers [38]. Advanced age, late diagnosis, inadequate diet, the prevalence of associated chronic conditions, and lack of self-care are related to chronic wound occurrence. Independent of the origin, the probability of wound occurrence doubles in people over 60 years, increasing mobility restriction issues, cardiovascular diseases, and diabetes, which are significant predictors of chronic wound development [39].

### 2.2. Healing Process Phases

After an injury, many biochemical events are activated to restore the damaged tissue and promote its healing [28]. The tissue repair process is divided into four phases, that are not very distinct, and overlap in time. Hemostasis, inflammatory phase, granulation tissue formation with matrix deposition extracellular (collagen, elastin, and reticular fibers), and remodeling are the stages of wound healing [28]. Overall, healing is a process in which damaged tissues are replaced by connective vascularized tissues, whether the injury was traumatic or necrotic, aiming at restoring tissue homeostasis.

Immediately after a skin injury, the hemostasis initiates to stop the bleeding, which depends on platelet activity, coagulation cascade, blood vessel constriction, and fibrin clot formation (Figure 1A) [12]. The clot formed acts in coaptating the wound’s edges, minimizing blood and fluid loss, protecting the organism against penetration of exogenous agents, and providing a provisional matrix for the beginning of the wound organization. The macrophages and neutrophils are recruited and secrete signaling factors, which govern the next stages of the tissue repair process [12].

Inflammation is mainly characterized by the presence of inflammatory cells in scar tissue [40] (Figure 1B). Overall, enhanced cell migration through venules and extravasation of serum molecules, antibodies, complement, and other proteins through the capillaries are important features of the inflammatory process [4]. During this phase, growth factors and cytokines are released around the injured site to attract immune cells, such as neutrophils and macrophages, and non-immune cells [40]. 

Neutrophils are the first cells to infiltrate the lesion followed by monocytes aiming at defending the tissue against pathogens. After reaching the injured site, monocytes are activated into macrophages, whose main functions are to kill the remaining microorganisms and remove tissue debris [41]. Macrophages are fundamental for the transition to the proliferative phase because they have a pivotal role in finishing debridement and secreting cytokines and growth factors for stimulating angiogenesis and extracellular matrix synthesis, and epithelialization [40]. The shift from the inflammatory phase to the proliferative phase occurs given the entry of fibroblasts into the wound, collagen accumulation, and the migration and formation of new endothelial structures inside the wound [41]. In this sense, the inflammatory process has direct effects on normal and abnormal wound healing, being potentially hazardous in chronic conditions and requiring adequate treatment [42]. In this regard, anti-inflammatory drugs have been explored as therapeutic agents in balancing inflammatory tonus during wound healing and providing pain relief [43]. However, there is evidence that nonsteroidal anti-inflammatory drugs (NSAIDs) delay both epithelialization and angiogenesis in the early phases of wound healing due to an antiproliferative effect [42,44]. Contrasting these data, some studies suggested that the use of ketoprofen and dexketoprofen positively influenced the healing process [45,46]. As expected, the association of these drugs with nanocarriers improved the anti-inflammatory action [47,48,49,50] and mitigated the adverse effects of NSAIDs, but additional studies are required for deep comprehension of the overall impact on the wound healing process. Thus, NSAIDs may be prescribed for post-soft-tissue injury or post-surgery to assist with pain control management and diminish inflammation; however, due to their negative effects on wound healing, their use is controversial and requires caution.

The proliferation stage provides granulation tissue formation, extracellular matrix synthesis, and angiogenesis [28] (Figure 1C). The fibrinogen of the inflammatory exudate is transformed into fibrin, forming a network where the fibroblasts are deposited for multiplying and secreting the protein components of the tissue cicatricial [4]. The fibroblasts appear to produce a new matrix to sustain cell migration, which will be replaced by an extracellular matrix composed of collagen and is directly linked to the strength of tension in the wound during the healing process [28]. The granulation tissue is responsible for filling the wound below the scab, favoring wound protection by forming a protective barrier against infections and helping wound epithelialization [4]. Granulation tissues are composed of elastin, procollagen, hyaluronic acid, and proteoglycans, which stabilize a structural repair scaffold to allow vascular growth and connective tissue formation [28]. Another important process in the proliferative phase is epithelialization, which begins with the mobilization and migration of epidermal cells from the wound margins followed by cell proliferation [4]. In the end, epithelialization occurs, a step that will lead to wound closure, and that is initiated by the migration of epithelial cells (keratinocytes) from the wound margins.

The last stage is the remodeling phase and consists of wound closure, organized collagen deposition, re-epithelialization, and scar tissue formation [4] (Figure 1D). In this phase, type III collagen is replaced by type I collagen, which has thicker, more resistant and more organized fibers, increasing the tensile strength of the lesion [41]. With the process of evolution, the reparative elements of wound healing are transformed into mature tissue with well-differentiated characteristics [28]. Lastly, collagen deposition is accentuated and most cells disappear, followed by the apoptosis of fibroblasts and endothelial cells, and scar tissue forming [41]. Chronic wounds can occur due to failure in any of the stages of the healing process (Figure 1E,F), which may be related to different factors such as pathogens (Figure 1G,H), circulatory diseases, diabetes, and metabolic disorders that may even lead to necrosis and mortality [27].

## 3. Soft Nanoparticles and Hydrogels for Wound Healing

Considering intact skin, the stratum corneum is the first cutaneous barrier, blocking the access of potentially harmful substances/stimuli, but hindering the absorption and/or percutaneous permeation of drugs as well. In cases of cutaneous lesions, the skin barrier is injured, which may influence drug penetration through cutaneous tissue [51]. In this regard, wound management becomes a challenge, mainly in cases of chronic lesions that do not progress through an orderly and timely process in promoting anatomical and functional integrity [52]. Therefore, the use of topical products has great value, since the physical protection and drug delivery at the wound site can accelerate the healing [10]. Generally, the device for wound dressing is selected based on the type, depth, extent, and location of the wound [52]. There is a variety of products on the market for this purpose, including traditional dressings, such as bandages, cohesive wraps, composite and non-adherent dressings, and the most complex dressings, which act in the various phases of the healing process [53]. Traditional dressings present some drawbacks, including their nonbiodegradability, high contact with recently grown granulation tissues, causing pain when removed, and infection susceptibility [54]. 

Improved wound dressings present multifunctional properties, including physical protection and wound moisture maintenance [54]. Among them, hydrogels stand out given their potential benefic properties for wound healing, for instance, biocompatibility, swelling reduction, tunable physicochemical profile, and non-toxic character [16]. Of particular importance, hydrogels emerged as a new type of wet wound dressing that can absorb or retain large amounts of water or biological fluids. Moreover, they present good biocompatibility, less susceptibility to infections and discomfort to patients, and an internal three-dimensional network porous structure that mimics the extracellular matrix, being an ideal scaffold for tissue regeneration [15]. 

Despite the abovementioned positive outcomes of hydrogels, there are some disadvantages, such as low mechanical strength, limited bioadhesiveness, high brittleness, poor drug permeation even in damaged skin, and adverse effects of drugs [17]. Additionally, conventional topical skin preparations are not adequate for incorporating unstable active pharmaceutical ingredients given the exposure to external stimuli and enzymatic degradation [51]. 

Then, combining hydrogels with strategies to improve the above shortcomings is required. In this sense, nanotechnology has sparked scientific and pharmaceutical interest given its biomedical applications [55]. Of particular importance, great efforts have been invested in preparing colloidal carriers to target and deliver drugs to a higher extent in the diseased sites [22]. The incorporation of nanoparticles in hydrogels became a promising approach given the high surface-area-to-volume ratio and ease of carrier delivery through the skin, reaching the desired layer but not the systemic circulation [51,56] (Figure 2). Thus, the idea of developing nano-based hydrogels in which nanofeatures are incorporated to modulate these properties has drawn special attention. 

Nanostructures are generally classified as hard and soft systems and both can be incorporated into a hydrogel matrix, yielding nano-based hydrogels with tailored properties [20,56]. Hard nanoparticles are produced using inorganic materials, such as gold, silica, graphene sheets, and carbon nanotubes. Despite their intrinsic functionalities, stability, and highly ordered structure, they can cause toxic effects upon their accumulation in tissues, triggering inflammatory reactions, and increasing the risk of tumor development [57,58,59]. On the other hand, soft nanoparticles (Figure 2) are considered more biocompatible in comparison to their hard counterparts, including liposomes (LP), nanocapsules (NC), polymeric nanoparticles (PN), nanospheres (NS), nanoemulsions (NE), solid lipid nanoparticles (SLN), nanostructured lipid carriers, nanogels, micelles (MC), and nanoparticles complexed with natural polymers [20,56]. Therefore, for biomedical purposes, the use of soft nanocarriers is more suitable and promising than the use of hard nanoparticles [60]. 

Considering the restrictions assigned to hydrogel, this section will cover recent findings regarding the association of this wound dressing with soft nanoparticles. The loading of drugs into a nanocarrier is advantageous due to the protection of drugs against tissue-native enzyme action, hydrolysis, oxidative inactivation, and decomposition reactions caused by substances from the external environment. Nanoencapsulation can also improve the photostability, solubility, and bioavailability of active molecules permeated through the skin from the formulation. Moreover, given the slow erosion of the nanostructure matrix in which the drug is entrapped, a prolonged drug release profile can be reached, enabling a reduction in the drug concentration in the formulation, and its administration frequency and toxicity. Overall, the drug-controlled release improves safety by allowing the maintenance of the concentration within the therapeutic window, which can also be enhanced by coating the surface of nanoparticles as a vectorization strategy to achieve target-specific therapies [51]. Lastly, improved mechanical properties and adhesion of formulations are achieved by preparing nano-based hydrogels [25].

Studies contemplating the incorporation of NE, LP, SLN, and polymeric systems in hydrogels (Table 1), were characterized and discussed herein. In general, the presence of the nanostructure in hydrogels enhanced the drug healing properties, mainly due to controlled release and the occlusion effect. In Table 2, we compiled the main findings of the studies covered herein, comparing the effects of hydrogels containing nano-based formulations, non-encapsulated drugs, and commercial products. In the following sections, the advantages of nano-based hydrogels are discussed.

### 3.1. Nanoemulsions

In the healing process, the lipid packing contained in the stratum corneum layer is rearranged, which eases the drug permeability through the tissue. Nanocarriers containing lipid components may have significant physiological advantages because of their similarity with the lipid bilayers in the stratum corneum [94]. Furthermore, the adhesiveness of lipid-based nanoparticles can favor the formation of an occlusive film in the applied area, preventing trans-epidermal water loss and microbial contamination, and contributing to skin repair [51]. 

Differently from the conventional emulsion, NE is referred to as dispersed systems with ≤100 nm droplets, high thermodynamics, and stability properties [95,96]. Classical emulsions are characterized by a coarse droplet size that can reach 1 μm. In comparison to conventional emulsions, the nanosized droplets are more kinetically stable, resulting in a lower tendency to creaming, sedimentation, flocculation, or coalescence [87,88]. Likewise, NE is composed of immiscible liquids that formed a single phase by an emulsifier and can be prepared by using materials from natural sources, such as essential oils and herbal-based compounds [97,98]. 

#### 3.1.1. Natural Compounds-Based NE

Natural products present several medicinal properties for wound management, e.g., antimicrobial, anti-inflammatory, analgesic, and antioxidant properties. However, they are still rarely used for treating skin lesions due to inadequate molecular size and low permeation of natural substances through skin layers, and physicochemical instability [98].

In this sense, their association with NE can better explore the advantages of natural-derived substances in the wound-healing process [97]. Nanoencapsulation of these therapeutics provides controlling their delivery to the injured side, enhancing their chemical activity, and increasing their stability [98]. Indeed, different studies have already demonstrated positive outcomes of using NE as dressings in some phases of wound healing (epithelialization, granulation, and collagen deposition) [99], which can be associated with the film formation on injured sites by the nano-droplets, favoring the formulation’s performance [98]. 

The integrity of droplet size after converting a NE formulation into a hydrogel is essential to maintain the advantages associated with the nanometric size, such as improved permeation/penetration in skin, controlled drug delivery, and enhanced pharmacological actions. In this sense, after associating the NE with the hydrogel, the physicochemical analysis performed in β-caryophyllene formulations demonstrated that the droplets were not affected after hydrogel preparation. Isoflavone aglycones-rich fraction-loaded NE incorporated into Carbopol^®^ Ultrez-21 hydrogel presented any significant differences between liquid and semisolid forms in mean size and polydispersity index, indicating that the droplets’ integrity was maintained. In contrast, chitosan hydrogels containing phenytoin associated with NE or NC showed particles in the micrometric range (>1 mm), probably due to the nanoparticle aggregation in the polymer network [71,100]. Eucalyptol NE also presented an increase in mean droplet size when converted into a Carbopol^®^ nanoemulgel (112 ± 0.77 and 139 ± 5.8 mm for NE and nanoemulgel, respectively) [87].

An essential feature of wound healing dressings is bioadhesion. This property describes the attachment of a system to the designated biological location, which can be modulated by the skin integrity, polymer nature, and type of nanostructure [71,75]. In addition, the gelling agent can contribute to the formulation adhesion. The hydroxyethylcellulose hydrogels containing β-caryophyllene-loaded NE presented 892.40 ± 81.53 mN and 444.37 ± 78.82 mN of maximum force for detachment in intact skin and injured skin (stratum corneum removal), respectively [75]. Chitosan hydrogels containing NC or NE of phenytoin presented 10.2 ± 0.7 mN.mm and 9.3 ± 1.0 mN.mm of the work of adhesion, respectively [71]. While hydroxyethylcellulose is a nonionic polymer that forms hydrogels with a wide range of viscosities, chitosan has a cationic nature, which can explain the higher bioadhesion, once positively charged groups interact with the anionic groups of the skin surface [71,100,101]. Formulations with high bioadhesion potential are not always desired because their removal may cause pain and secondary damage to tissues under regeneration [90,102]. Remarkably, in a diabetic wound model, chitosan-based hydrogel encapsulating perfluorocarbon emulsions, epidermal growth factor (EGF)-loaded chitosan nanoparticles, and polyhexa-methylene biguanide (PHMB) was less adhesive than gauze and a commercial wound dressing, mainly due to its moist character, which is a highly advantageous feature for wound care [88]. 

Poorly water-soluble drugs may be encapsulated into a lipophilic environment of an oil droplet in a nano dimension and stabilized through an optimized amount of surfactant/co-surfactant mixture along with an aqueous phase, enhancing the drug skin permeability. The skin permeation profile showed that the thymoquinone (TMQ)-loaded-NE increased the cumulative drug permeation (549.16 ± 3.10 µg/cm^2^) in comparison to non-encapsulated TMQ (120.75 ± 2.43 µg/cm^2^). This finding is probably due to the presence of surfactants in NE composition [103,104]. In addition, the skin permeation of the TMQ-NE formulation (965.65 ± 12.84 µg/cm^2^) was significantly higher than the TMQ-gel formulation (150.93 ± 1.80 µg/cm^2^). The deeper penetrability of TMQ from TMQ-NE hydrogel provided the formation of an extensive and organized cellular structure in healed skin tissues compared to the groups that received no treatment and non-encapsulated TMQ. 

Similarly, permeation studies demonstrated that the NE-based hydrogel formulation promoted higher retention of isoflavones in the stratum corneum and dermis than the liquid form of NE, probably because of hydration of the most external skin layer [77,105]. After removing the stratum corneum, superior quantities of isoflavones were found in the epidermis, dermis, and acceptor fluid, indicating a penetration enhancement. In addition, the cumulative quantity of curcumin permeated was 773.82 ± 1.08 from nano-based hydrogel and 156.90 ± 0.95 µg/cm^2^ from the non-encapsulated curcumin gel [85]. The presence of polysorbate 80 in formulations could enhance drug dermal delivery by influencing the *stratum corneum* layer hindrance and intracellular lipid barrier, increasing the skin-permeation and deposition of drugs [103].

Plant-derived essential oils and their bioactive compounds present pharmacological properties that are relevant for the wound healing process, such as antifungal, antibacterial, antioxidant, and anti-inflammatory [87,106]. Eucalyptol (EU) is the most important component of Eucalyptus and could improve the wound-healing process by enhancing the capillary permeability after its administration via the intradermal route. Table 3 demonstrates the percentage of wound contraction for Eucalyptol in an excisional wound model. The optimized formulation had a healing rate similar to the commercial product (Quench^®^ cream containing silver sulphadiazine), reaching 100% of wound contraction after 15 days. Such an effect was attributed to the increased contact time with the skin provided by the nanocarriers, which enhances EU permeability and the final pharmacological effect [87].

An adequate viscosity and spreadability factor are essential for formulations that will be applied to the wounds, mainly because these properties could interfere with the wound site coverage area, drug distribution, and application-related pain. The presence of isoflavone aglycones-rich fraction-loaded NE in a Carbopol^®^ Ultrez-21 hydrogel presented a reduction in viscosity when increasing the shear rate, assigning to the formulation a non-Newtonian pseudoplastic behavior described by Oswald’s model [77]. A similar profile was observed for Carbopol^®^ 940 hydrogel containing curcumin-loaded NE and TQM-loaded NE [84,85]. Nanosystems with lipid components and these common gelling agents usually present such behavior [64,89], which could be interesting for painful wound management once the formulation begins to flow with a minimum applied force.

Curcumin is a widely studied natural bioactive compound that has remarkable actions, including anti-inflammatory, antioxidant, and anti-infective effects. Additionally, curcumin hastens the healing process by acting over the different stages of the wound healing process [84]. However, low aqueous solubility and skin penetrability hinder its therapeutic use for skin disorders [85,107]. Algahtani and co-workers prepared a curcumin-loaded NE formulation by the high-energy ultrasonic technique and incorporated it into a Carbopol^®^ 940 hydrogel for topical application in cutaneous open wounds [85]. In the animal model of wound healing, hydrogel containing curcumin-loaded NE promoted a faster epithelization than hydrogel of non-encapsulated curcumin and commercial formulation (10, 14, and 11 days, respectively) (Figure 3). Furthermore, histopathological analysis revealed a high amount of granulomatous mass, collagen fibers, and a reduced number of inflammatory cells for the animals treated with a hydrogel containing encapsulated curcumin [85]. 

Some studies presented the development of hydrogels composed by NE or NC to compare these systems in the wound healing process. Nanocapsules are reservoir systems, where a polymer surrounds an oil core, while NE did not present any polymer in composition [108]. Aiming at studying the *Melaleuca alternifolia* essential oil (tea tree oil-TTO) skin protective effect against UVB radiation and cutaneous wound healing potential, Flores and co-workers prepared TTO-loaded NE (spontaneous emulsification method) and NC (interfacial deposition of preformed polymer) [109] and further incorporated these systems in hydrogels [65]. The Carbopol^®^ Ultrez was used as a gelling agent, which provides satisfactory properties for topical application. Carbomers are polymers derived from acrylic acid, of high molecular weight, and dispersible in water. The high concentration of the carboxyl group in the Carbopol ionizes more at a higher pH and leads to repulsion between charges of the same group promoting a swelling property to the hydrogel, which is desirable for wound exudate absorption [110,111]. Formulations had pH values compatible with the skin (acid range), a plastic rheological behavior, and good spreadability. Notably, hydrogels containing nanostructures lead to a decrease in the spreadability factor (~4.5 mm^2^/g), in comparison with the hydrogel vehicle (~9.0 mm^2^/g), suggesting that nano-based formulations could be more distributed per application area [65]. Otherwise, hydrogel containing non-encapsulated TTO presented a lower spreadability factor (~2.85 mm^2^/g), demonstrating that this formulation needs a higher force to be spread, which is a disadvantage in painful wounds. Despite that, hydrogel containing NE presented a significative effect in incisional wound healing evaluation, the NC-based semisolid was superior. A plausible reason to explain these advantages is that through steric and electrostatic mechanisms the polymeric wall could provide superior adsorption of the nanoparticles at the tissue surface, improving the biological effect of formulations.

Reactive species are secreted by defense cells throughout the inflammatory phase of wound healing aiming to protect the tissue against microorganisms and stimulate angiogenesis. Despite these functions, the excess of reactive species can create an imbalance in the oxidative tonus, promoting macromolecule oxidation and damaging cellular structures, which impair wound healing [112]. As expected, NC-based hydrogel improved antioxidant defenses, which may be because of the protective function of the polymeric wall against terpenes volatilization. 

Conversely, Cardoso and co-workers obtained distinct findings. The study developed chitosan hydrogels containing phenytoin associated with NC or NE that presented similar healing potential in a model of rat dorsal skin incision [71]. Notably, NC without the drug also presented a significant healing potential when compared to the positive control, suggesting that the hydrogel composition may contribute to the pharmacological effect. In this sense, antibacterial and antioxidant properties were already assigned to chitosan (gelling agent) and the NC polymer [113,114]. The main differences were observed in physicochemical parameters in which the NE presented a higher encapsulation efficiency, lower mean size, and zeta potential than NC. These characteristics may be linked to the polymer (Poly-epsilon-caprolactone—PCL), which interferes with organic phase viscosity and presented terminal carboxylic acid in its chemical composition. In addition, a different drug release profile was observed for the NC and NE-based hydrogels; NC provided greater control over drug release. Confronting the nanocarriers release profile, the authors concluded that phenytoin has first to diffuse through the nanostructure and subsequently through the hydrogel network to reach the medium, as previously demonstrated in the literature [71]. 

The nanostructure composition and molecular organization can modulate the profile of skin drug permeation, consequently impacting the wound-healing process [71]. Additionally, impairment of skin integrity could modify drug permeation/penetration, once in wounds and injuries the cutaneous barrier properties may vary [71]. Concerning this topic, phenytoin permeation and penetration in skin layers were investigated using uninjured and injured skin. Regardless of the formulation (liquid or semisolid), neither the NE nor the NC reached the receiving medium after application to intact skin. 

Diabetic patients usually have impaired metabolism, immune system deficiency, and high susceptibility to recurrent infections. These factors impact the healing rate of diabetic skin wounds, such as foot ulcers [38]. Animal models of diabetic wounds are based on disease induction, which is usually with streptozotocin, followed by wound generation in the animal’s dorse [69,88]. Considering the urgent improvements required in diabetic wound management, Lee and Lin showed a chitosan-based hydrogel associating perfluorocarbon (PFC)-loaded NE, epidermal growth factor (EGF)-loaded chitosan nanoparticles, and polyhexamethylene biguanide (PHMB) named P_E_E_NP_PCH as a new approach for diabetic wound healing. The individual encapsulation of PFC and EGF might prevent their interactions with PHMB and with each other before their release to the wounded site. PFC-loaded NE was produced by the homogenization method and epidermal growth factor nanoparticles by coacervation. Such particles were further mixed with the PHMB given its antimicrobial properties. Hydrogels were fabricated using a modified freeze-thaw cycling method forming a moist semitransparent membrane with a porous morphology in both surface and internal structures. Scanning electron micrograph demonstrated that several particles were adhered to the fibrous surface and/or entrapped inside the fibers throughout the hydrogel matrix. According to the PHMB concentration and the number of particles, some parameters of the hydrogels were affected. The stiffness was increased and the tensile strength (stress) and elasticity (strain) of hydrogels were reduced as the number of nanoparticles increased [88]. 

Growth factors associated with hydrogels can significantly enhance the regenerative rate of wounds facilitating tissue regeneration and growth. Furthermore, hydrogels prepared with different soft nanoparticles could potentially enable the organization of cells during tissue maturation, improving the healing potential of devices by modulating the cellular fate in wounds [56]. The chitosan-based heterogeneous composite hydrogel containing the EGF may facilitate keratinocyte proliferation, which is favorable for tissue regeneration. In addition, the formulation presented in vitro antibacterial and anti-inflammation actions and oxygen delivery, which are desirable properties of diabetic wound healing devices. Corroborating these arguments, the in vivo investigations demonstrated that the nano-based semisolid formulation quickly healed the wound by displaying the smallest visible size among all groups, providing re-epithelialization and collagen deposition as well as reducing inflammation over the treatment [88]. 

#### 3.1.2. Synthetic Drugs-Based Nanoemulsions

Synthetic compounds also present advantages when nanoemulsified and incorporated into hydrogels as demonstrated by Ansari and co-workers [86]. Crisaborole (CRB) is a (5-hydroxy-1,3-dihydro-2,1-benzoxaborole) phosphodiesterase (PDE4) inhibitor indicated to treat moderate atopic dermatitis [115]. This study aimed to incorporate CRB-loaded NE into a chitosan hydrogel for providing prolonged drug delivery and enhanced penetration in wounds. Both NE and chitosan hydrogel enhanced the CRB release over 24 h (65.65 ± 3.7 and 74.45 ± 5.4% for NE and hydrogel, respectively), which was considered an advantage since CRB is a hydrophobic drug and presents a limited release from simple gels. In the animal model of an excision wound, complete epithelialization was observed on the 20.72 ± 1.18 post-wounding day in the group treated with nano-based fusidic acid hydrogel, whereas the CRB-NE-loaded chitosan hydrogel demonstrated an effect on the 21.45 ± 1.46 post-wounding day [86]. 

### 3.2. Solid Lipid Nanoparticles

Solid lipid nanoparticles (SLNs) have gained significant attention in topical applications given their high stability, safety, and occlusive film-forming properties [116,117]. In addition, increased active solubility, protection against degradation, controlled and target release, and payload can be provided by incorporating drugs into SLN. These carriers are compatible for use on inflamed skin as well, because of the non-skin irritation potential of the lipid matrix [89]. The SLNs also have a relevant physiological advantage given the similarity in their lipid composition with the lipid bilayers [116].

#### 3.2.1. Natural Products-Based Solid Lipid Nanoparticles

As aforementioned, herbal products have gained attention in the wound healing process. Astragaloside IV is the majority compound present in Astragali Radix (AR) (the root of *Astragalus membranaceus*), a widely used herb for enhancing tissue and organ repair and recovery. This compound was incorporated into SLN, using glycerol tristearate as a lipid component, and further thickened in a Carbopol^®^ 934 hydrogel. The size of the astragaloside IV-based SLNs in the carbomer hydrogel was 394 nm, whereas that of the astragaloside IV-based SLNs was 318 nm. These results demonstrate that the nanoparticulate structure was maintained after incorporating the nanocarriers in the hydrogel. Conversely, simvastatin-loaded SLN produced by a hot high-pressure homogenization method and also mixed with a hydrogel scaffold of Carbopol^®^ 934 presented an average particle size of 294 nm when determined while the system was warm and within 24 h of preparation. However, after seven days of hydrogel fabrication, the size increased (>1000 nm), which can be due to the presence of the gelling matrix. 

To assess the in vitro wound healing effect of astragaloside IV, migration was monitored. The astragaloside IV-based SLNs increased the wound closure of keratinocytes in comparison to the solution, at the same concentration after 48 h of incubation [64]. These data showed that astragaloside IV-based SLNs enhanced the proliferation and migration of keratinocytes, indicating its capability to accelerate wound re-epithelialization. In the in vivo astragaloside IV solution, astragaloside IV-based SLN-gel, and blank SLN-gel were administrated every two days. Corroborating the in vitro findings, Figure 4A demonstrated that the nano-based hydrogel contributed significantly to wound repair when compared to other groups on 4- and 12-days post-wounding. The topical treatment also contributed to angiogenesis and newly formed collagen fiber deposition (Figure 4B). In addition, many blood vessels were found in the regenerated skin of the wounds treated with astragaloside IV solution and astragaloside IV-based SLN-gel (Figure 4C) at 3- and 7-weeks post-wounding. As shown in Figure 4D, CD31 immunohistological staining revealed no angiogenesis in the vehicle group. In the blank SLN-gel group, some tiny blood vessels were observed. Then, the astragaloside IV solution and astragaloside IV-based SLN-gel could consistently contribute to the angiogenetic effect by enhancing the quantity and size of blood vessels [64]. 

As previously mentioned, the therapeutic translation of curcumin is hindered due to its poor water solubility, low permeability, and physicochemical stability [107]. The pro-oxidant effects at high concentrations also limit curcumin use, resulting in concentration-dependent cytotoxic, genotoxic, apoptotic, and ROS-generating effects [118]. In this regard, Sandhu and co-workers developed curcumin-loaded SLN using Compritol^®^ 888 ATO and Phospholipon 90G (soya lecithin) as a lipid matrix [82]. Tetrahydrocurcumin (THC), a metabolite of curcumin, was incorporated into SLN of the same composition by Kakkar and co-workers [68]. Despite its higher anti-inflammatory effect in comparison to curcumin, this compound presents poor aqueous solubility (0.0056 mg/mL) and a log *p* value of 2.98, suggesting that THC may present limited skin permeation [68]. 

After preparation, both SLNs were incorporated into Carbopol^®^ 934 hydrogels for further investigation. The SLN association with hydrogel extended the curcumin release for up to 120 h (81.95 ± 0.64%), fitting on the zero-order model, which describes a drug release at a constant rate. Following, a nitrogen mustard-induced burn and excisional wound models were used to investigate the wound-healing effect of curcumin-loaded SLN hydrogel [82]. A significant reduction (184%) in inflammation was observed in the groups treated with hydrogel in comparison to the induced group. The pharmacological effect was attributed to the modulation of interleukin 1 (IL-1) and TNF-α expression by curcumin, which present important roles in the inflammatory response modulation. In the excisional model, on day 11, hydrogel resulted in the complete wound closure (95.76 ± 7.85% reduction in wound area), which was superior to the positive control (57.85 ± 8.85%), blank SLN hydrogel (81.18 ± 2.66%) and non-encapsulated curcumin hydrogel (74.75 ± 3.53%), as elicited in Figure 5. 

Interestingly, the blank formulation presented healing properties due to the occlusive nature of SLNs, maintaining moisture at the wound mainly by the phosphatidylcholine presence. The same effect was demonstrated in hydrogels containing simvastatin-loaded SLN and Poloxamer 188 in composition, which has wound healing properties [89,119] as well as for hydrogels containing Cu (II) Schiff base 8-hydroxy quinoline complex-loaded SLN using Tween 80 as surfactant [92]. 

Concerning the oxidative stress in the wound environment, the nano-based hydrogel was superior in minimizing damage in comparison to free curcumin hydrogel. These findings demonstrated that the nanoparticles could improve the drug availability, stability, permeability, and bioavailability at the wound bed given their encapsulation into the lipid core. In addition, the formulation also increased the vascular endothelial growth factor levels, which is the most important proangiogenic mediator. 

The percentage of water loss depends on the potential of the formulation to form an occlusive layer on the skin. The in vivo wound healing potential of the THC-SLNs hydrogel was assessed using the excision wound mice model. The reduction in wound sizes was higher when THC-SLNs hydrogel was applied, maintaining an intact epidermis and normal dermis with no signs of inflammation, increasing the collagenous mass at the injury site and modulating the tissue oxidative state [68]. 

#### 3.2.2. Synthetic Drugs-Based Solid Lipid Nanoparticles 

Gupta and co-workers presented self-gelling SLN-hydrogel containing simvastatin as the wound-healing drug [89]. The wound-healing potential of the simvastatin-loaded SLN hydrogel was assessed in an excisional model using rats. The nano-based hydrogel has a wound closure potential 5- and 10-fold higher in comparison to the groups that received no treatment and commercial preparation. In addition, histological analysis demonstrated a decrease in inflammation, an increase in collagenous mass, and progressive changes in the epidermal and dermal region, including keratinization and full-thickness epidermal regeneration [89]. 

The presence of bacteria in wounds can delay the healing process and even lead to systemic infection and death at worst [90]. Opportunistic bacteria easily colonize the wound site and create a protective biofilm, hindering the infection cure [120]. Given this, topical application of antimicrobials can facilitate the healing process, however, many multi-drug resistant bacteria strains have already been identified [121]. Nisin is a positively-charged antibacterial peptide, which presents effective inhibitory action against a broad range of microorganisms. Despite the advantages, antibacterial peptides require strategies for preventing their degradation or inactivation, enhancing the biological action, and providing release control at the wound site [90]. Thus, purified nisin was associated with SLN and incorporated into a hydrogel scaffold composed of gellan gum and a mixture of gellan gum and alginate. Stearic acid-based SLNs were manufactured using the double emulsification/solvent evaporation technique. The hydrogel behavior in a humid environment was evaluated to simulate the exudate, which is commonly present in infected wounds. After 1 h, all the samples presented a swelling ratio between 2520 ± 40% (GG + NP_NSN) and 2960 ± 150% (GG/Alg + NSN) in comparison to their initial mass. The antibacterial efficacy was tested in contact with *S. pyogenes*. The size of inhibition zones was bigger for both hydrogels containing the SLN than hydrogels containing non-encapsulated nisin. Moreover, this study also presented a wound-healing assay to evaluate the influence of sample extracts on cell migration and wound healing. Hydrogel composed only of gellan gum and nisin-SLN presented a wound closure percentage similar to the control group (more than 80% closure). On the other hand, the mixture of gellan gum and alginate seems to be less efficient, presenting a wound closure percent of 31.4 ± 6.2%. No cell adhesion was observed, which may be of great importance for devices intending wound healing applications since changing them would not cause pain or damage to the regenerated skin [90,102]. 

Natural polysaccharides have been commonly used materials for the fabrication of novel wound dressings due to their excellent biocompatibility, high swelling capacity, low cost, and presence of numerous functional groups available for prospective modifications [90]. Among them, alginate gained increased interest because of its biocompatible and gelling properties [90]. Likewise, natural gums have been extensively studied as thickening agents for nanosystems due to their ability to form viscous dispersions at low concentrations [122]. 

El-ezz and co-workers developed a formulation containing Cu (II) Schiff base 8-hydroxy quinoline complex (CuSQ)-loaded SLN hydrogel and investigated its wound healing potential in an excision wound model in rats. Quinoline compounds have already demonstrated pharmacological effects, including antibacterial, antioxidant, and anti-inflammatory [92]. The SLN formulation was produced by water/oil/water type double emulsification method, using soy lecithin and cholesterol as the lipid matrix. After physicochemical characterization, the nanoparticles were gelling with Carbopol^®^ 940. The CuSQ release profile from hydrogels demonstrated a rapid initial release followed by a slower rate. The burst effect can be associated with drug desorption in the particle surface, while the controlled release was attributed to the drug solubilization into the lipid matrix, requiring drug partition from the matrix. Wound healing was faster at both low (0.05 ug/mL) and high concentrations (0.2 ug/mL) of CuSQ-treated groups as compared to the control and standard (0.1% Garamycin cream) groups. On the 12th day post-wounding, almost 100% of healing was reached by CuSQ-SLN hydrogels. More efficient wound healing and closure were observed in the hydrogel containing the highest concentration of CuSQ-SLN, where a newly developed epithelium with keratinization wrapped the wound site.

### 3.3. Liposomes

One of the major concerns of current topical treatment is that most drugs only penetrate the skin to a limited extent. To overcome this issue, the incorporation of drugs into vesicular lipid delivery systems, such as LP, is an approach being investigated [123]. They are vesicles comprised of concentric phospholipid bilayers, separated by aqueous compartments, which have been described as topical drug carriers. In general, liposomal drug formulations are seen to be more effective and better tolerated than corresponding conventional pharmaceutical formulations, highlighting the potential of liposomes as cutaneous drug delivery devices for wound healing [124].

The combination of anti-infectious therapy and healing-promoting moisturization is promising in cutaneous wound management. Most of the available commercial products intended to support tissue repair lack anti-infective properties and are even contra-indicated to treat infections [125]. Furthermore, some antimicrobial drugs cause the inhibition of granulation and epithelization, hindering the healing process [12]. In this sense, Reimer and co-workers [61] prepared a polyacrylic acid hydrogel containing povidone-iodine-loaded phosphatidylcholine LP for achieving both an antiseptic and moist treatment. The authors performed a clinical evaluation using healthy patients with mesh grafts as a suitable and very sensitive model for the assessment of efficacy and tolerability of preparation for wound treatment, consisting of a proof-of-concept phase II (liposomal povidone-iodine hydrogel vs. Bactigras). The findings demonstrated that the rate of neo-epithelization was significantly greater in the group treated with nano-based formulation than in the Bactigras. Then, the combination of the povidone-iodine with the good tolerability and drug delivery properties of LP prevented infections and promoted wound healing. Indeed, LP has characteristics that are useful in their application as topical drug carriers such as site-specificity, water-binding capacity, drug release control, facilitated penetration and drugs’ release on skin and mucosa, and moisture retention [124,126].

The application of collagen mimetic peptide tethered vancomycin-loaded LP hybridized to collagen-based hydrogels for the treatment of methicillin-resistant *Staphylococcus aureus* (MRSA) infections was investigated by in vitro and in vivo approaches [78]. The authors prepared a hydrogel composed of collagen-fibrin co-gels via triple-helical integration with collagen, containing collagen mimetic peptides to stably hybridize vancomycin-loaded liposomal nanocarriers. Liposomes were formulated using a thin film hydration method with a lipid composition of 1,2-dipalmitoyl-sn-glycero-3-phosphocholine (DPPC):Cholesterol:1,2-distearoyl-snglycero-3-phosphoethanolamine-N-[methoxy(polyethylene glycol)-2000-maleimide](DSPE-PEG-Mal) at a molar ratio of 73:24:3. Conventional antimicrobial formulations are devoid of controlled active release, presenting a critical oscillation of drug concentrations at different time points and promoting antibacterial resistance and wound infection recurrence. The formulation enables control over vancomycin release for long periods and improves vancomycin safety by maintaining fibroblast cell viability in the wound bed [78]. Moreover, enhanced antibacterial action was observed for liposomal-based hydrogel in comparison to the conventional formulation against MRSA. The in vivo antibacterial activity of the lipo-loaded co-gels was evaluated using a mice model of a punch biopsy wound induced by luminescent MRSA multiple inoculations (day 0 and day 1). Corroborating the in vitro findings, the nano-based hydrogel was able to inhibit bacterial growth over the experimental evaluation, exemplifying the advantage of vancomycin-controlled release for prolonged periods [78]. Collagen mimetic peptide conjugated vancomycin-loaded LP formulation incorporated into a collagen-based scaffold presented antibacterial activity even after multiple inoculations (Figure 6). This effect can be attributed to the enhancement of the formulation residence time in the wound bed, promoting a controlled drug release and limiting bacterial growth following multiple infections during wound healing.

Thapa and co-workers (2021) studied the association of antibacterial peptides with a liposomal-based hydrogel for treating infected wounds [83]. Bacteriocins are bacterial-derived peptides that present antimicrobial properties and arise as viable alternatives to antibiotics owing to their natural origin and low propensity for resistance development [127]. For better exploring the potential applications of bacteriocins, phosphatidylcholine LP was prepared using the thin film hydration technique. The hybrid hydrogel was composed of Pluronic F127 (PF127), bacteriocin Garvicin KS (GarKS), and ethylenediaminetetraacetic acid (EDTA)-loaded LP and glutathione. Given the hydrophilic nature of the hydrogel scaffold, a high swelling percentage was obtained, which is desirable for wound exudate absorption. This property can promote fibroblast proliferation and migration, and homeostasis rate after bleeding by enabling the clotting factor concentration [128,129]. Remarkably, wound infections are a mixture of Gram-positive and Gram-negative bacteria, then a suitable antibacterial formulation composition for both bacterial types is required for effectively treating infected wounds. Corroborating the in vitro finding, the mouse model of infected wounds demonstrated potent antibacterial effects of nano-based hydrogel, providing proof-of-concept for the successful development of GarKS-loaded LP topical formulation for effective treatment of wound infections.

Wounds caused by cutaneous leishmaniasis (CL) are difficult to manage [35]. Leishmaniasis is an orphan tropical disease caused by Leishmania parasites [36], and depending on the parasite species and the immune state of the host, leishmaniasis progress may vary [35]. The current pharmacological treatments present toxic issues and topical dosage forms are rarely addressed for CL management [36]. Miltefosine is the only FDA-approved drug to treat CL via the oral route, but it has significant adverse effects [7]. Then, a topical treatment would have the great advantage of minimizing the systemic circulation of drugs and general toxic issues. Of particular importance, reaching the dermal amastigotes is of the utmost importance for healing an infection, which can be improved by loading drugs into nanocarriers, including LP. In this context, Peralta and co-workers (2022) designed miltefosine-LP formulations and investigated in vitro efficacy, cytotoxicity, and in vivo irritation and efficacy in an animal model of CL caused by *L. amazonensis* [7]. Liposomes were prepared by the thin-film method and were composed of unsaturated phospholipids or a mixture of unsaturated phospholipids. Both systems presented particle size in the nanometric range, low cytotoxic effects on macrophages, and high inhibition of parasitic viability in comparison to non-encapsulated miltefosine. Additionally, the nanoencapsulation modulated the drug release profile. The Carbomer 934-P NF was used for preparing the hydrogel containing the LP, which presented a homogeneous aspect, pH at the acid range, and adequate spreadability properties. The in vivo model demonstrated that nano-based formulation enhanced the miltefosine anti-parasitic action, accelerating the clinical cure, and the complete elimination of parasites without causing systemic toxicity. The authors discussed that the phospholipid molecules used to produce the LP, modulate skin lipids and enhance nanocarrier permeation, therefore diminishing the barrier function of the skin, and acting as a penetration enhancer, leading to superior drug penetration.

Wounds, particularly burns, are prone to potentially life-threatening infections due to inadequate perfusion, resulting in hindered immune cell migration and antimicrobial agents’ delivery from conventional pharmaceutics [34]. Considering this issue, a research group investigated the potentialities of a chitosan hydrogel containing mupirocin-loaded phosphatidylcholine LP as an advanced delivery system for improving burn wound management [62]. The antibiofilm property and therapeutic suitability in a mouse model of burn were further investigated using the same nano-based formulation [63]. LP formulation was prepared following the dry film method and submitted to size reduction by sonication to be optimal for topical applications, namely between 200 and 300 nm. The liposomal suspensions showed an entrapment efficiency of 62.4% (±8.8%) and a mupirocin concentration in the liposomal suspension of 1.87 mg/mL. The optimized formulation was stable at different storage conditions (cold and accelerated studies) and was incorporated into chitosan hydrogel prepared by manual stirring (10% of final liposome concentration). The in vitro release studies on Franz diffusion cells demonstrated a prolonged and controlled release of mupirocin from the nano-based hydrogel. Corroborating this finding, the permeation study in pig skin followed a similar release pattern as observed in the in vitro study. In addition, a satisfactory antimicrobial potential was obtained for chitosan hydrogel containing mupirocin-loaded phosphatidylcholine liposomes [62] and an antibiofilm effect [63]. 

In this regard, chitosan has a known potential in burns treatments given its intrinsic antimicrobial properties combined with the potential to promote drainage and gas exchange and prevent exudates buildup [130]. Concerning safety evaluation, the liposomal formulation was noncytotoxic (0–10%) or moderately cytotoxic (>10–20%) to HaCaT cells [63]. Remarkably, other agents that are widely used in wound dressings, such as silver, can exhibit significant toxic effects, reinforcing the advantages of associating drugs with nanocarriers. The formulation presented promising effects in a burn mouse model, where the group treated with chitosan hydrogel containing mupirocin-loaded phosphatidylcholine LP showed faster healing in comparison to the other treatments. Despite some limitations regarding the in vivo model, the findings showed that the system is equally adequate and safe for administration on the wound site. Overall, both studies demonstrated the advantages of associating drugs with nanocarriers for preparing hydrogels intended for wound healing.

The association of natural actives with nano-based carriers is a potential strategy to improve their biological properties and prospective novel application [98], including for developing wound healing devices. The possible quercetin application as a healing enhancer has been supported given the fibroblast proliferation, immune cell infiltration reduction, changes in fibrosis-associated signaling pathways, and minimized fibrosis and scar formation in wound healing [67]. Similarly, curcumin has been shown to possess relevant wound-healing properties, by improving granulation tissue formation, collagen deposition, tissue remodeling, and wound contraction [131]. Hydrogels containing LP of quercetin [67] or curcumin [91] were already investigated as potential healing enhancer candidates. The nanocarriers were composed of phosphatidylcholine and produced using the thin lipid film hydration method. Previous to LP incorporation, the hydrogel scaffolds were obtained by optimized gelatin to Carbopol^®^ ratio [67] or 2% Carbopol^®^ (*w*/*v*) [91]. The rheological behavior of the hydrogel containing curcumin-loaded LP demonstrated a non-Newtonian flow, which is a desirable feature to enable easy spread onto the skin. 

Concerning the release drug rate from the formulation, a typical biphasic release pattern was observed, in which a burst event was followed by a slower sustained release of quercetin [67]. Then, nano-based formulation provided an enhancement in the bioavailability and modulation in the delivery of quercetin at the wound site. Lastly, the in vivo performance of the formulation in wound healing was assessed using the full-thickness punch wound and surgical wound models. The data showed that hydrogels containing LP had the highest percentage of wound contraction, presenting epidermis and dermis layers with a normal physiological structure. Corroborating these findings, the histomorphometric values showed the highest percentage of collagen, lowest inflammatory rates, highest presence of microvessels, and reepithelization rates at the wound site in animals treated with nano-based formulation [91]. Given the great water adsorption, bioadhesive property, and sustained release of the actives from the nano-based hydrogel, the wound healing of rats was greatly enhanced in comparison to the other groups, suggesting that the association of phytochemical with LP can greatly promote the rate of wound healing.

One of the common diabetes complications is that wounds of diabetic patients tend to heal more slowly, resulting in undesirable outcomes [38]. In this regard, insulin has been studied as a promising therapeutic wound healing agent, including for topical uses. However, insulin faced major problems regarding stability and bioavailability after cutaneous application. Thus, an insulin-loaded liposomal mucoadhesive chitosan hydrogel was designed aimed at providing sustained release and extended stability for the peptide [70]. Dry thin film hydration was the technique chosen for the LP preparation, which was composed of egg phosphatidylcholine and cholesterol. The authors also conducted a clinical study for assessing the potential applications of the formulations in diabetic wound healing. The results showed an improvement of 16-fold in the healing rate in comparison to the control group that received the placebo wound healing dressing, with a significant reduction in ulcer erythema and no hypoglycemia. Thus, the authors showed that insulin-loaded liposomal chitosan gel showed is a highly stable and therapeutically promising drug delivery system for diabetic wound management. 

### 3.4. Polymeric Systems

#### 3.4.1. Nanocapsules and Nanospheres

Polymeric nanoparticles are carriers characterized by a biocompatible polymer. The structures can be organized in vesicular systems containing a solid polymeric shell surrounded by an oily core, where the drug can be dissolved in the core or/and adsorbed to the polymeric wall, named NC [24]. NS is based on a continuous polymeric network in which the drug can be retained inside or adsorbed onto the surface [108]. Due to some particular properties (e.g., particle shape and size, and polymer surface charge), NC and NS can facilitate the tissue permeation of actives resulting from intimate contact with the biological surfaces and particle deposition [108,132].

The polymer confers some advantages for hydrogels containing polymeric particles, such as prolonged drug release, enhanced stability, improved mechanical strength, and increased adhesion [17,108]. Given this, in this section, we compiled studies that explored the incorporation of soft nanoparticles and how they can modulate various properties of hydrogels concerning their application in wound dressing. 

In the scientific literature, some studies have already compared the healing potential of NC with other systems, such as NE [65,71]. The findings demonstrated a superior effect of polymeric particles, which was assigned to the steric and electrostatic mechanisms of the polymeric wall that could enhance the adsorption of the nanoparticles at the tissue surface, improving the final pharmacological biological effect [65,132]. Even presenting a similar healing rate, NC showed a more sustained release of drugs in comparison to NE [71]. The release of drugs from polymeric nanoparticle systems depends on physicochemical changes in the polymeric structure and may follow distinct mechanisms (Figure 7), such as I) drug diffusion through the matrix; II) polymer swelling and erosion, III) polymeric shell degradation, or IV) a combination of them [22,80,108,132].

Aliphatic polyesters and copolymeric counterparts are very common synthetic materials and have been studied for producing drug delivery systems given their biocompatibility and biodegradability. The most common polyesters are poly (lactic acid) (PLA), poly (lactic-co-glycolic acid) (PLGA), and poly(ε-caprolactone) (PCL). Compared with PLA and PLGA copolymers, PCL provides a longer degradation period [108,132]. In addition, PCL is an interesting versatile polymer that presents low toxicity, biocompatibility, biodegradability, and compatibility with a wide range of drugs, justifying its application in the search for new potential treatments, including the development of wound healing devices [132]. The PCL lipid-core NCs present a core structured by a polymeric wall surrounding a dispersion of solid lipid (sorbitan monostearate) and liquid lipid (caprylic/capric triglyceride). Pires and co-workers developed a *Caryocar brasiliense* oil-loaded PCL lipid-core NC (LNC_CBC_) and evaluated the formulation on the healing of wounds [76]. To provide a cationic surface to LNC_CBC_, another formulation was prepared by coating the particles with chitosan (LNC_CBC+_). Chitosan is a common natural polymer that has been broadly used as a drug carrier and gelling agent in hydrogels, presenting biocompatibility, endogenous degradable products of metabolization, and gelation and mucoadhesive properties [132]. Nanocarriers formed or covered by chitosan can present a cationic surface charge due to amino groups [80,132].

The NC suspensions were incorporated into a hydrogel using hydroxypropylmethylcellulose as a gelling agent (HG-LNC_CBC_ and HG-LNC_CBC+_). The anti-inflammatory action of the oil was confirmed by the absence of inflammatory exudate, swelling, and hyperemia in the treated area. Furthermore, the nano-based hydrogels formed a film in the biological surface that promoted physical protection, avoiding microorganisms’ penetration from the external environment and water and heat loss from the granulation tissue. Chitosan can act as a wound-healing catalyzer by accelerating the polynuclear cell infiltration in the injured area, stimulating macrophage migration and fibroblast proliferation. The cationic nature of chitosan may enhance interaction with cells, which supports the modulation of these cellular events [76,80,126]. Additionally, the *Caryocar brasiliense* oil nanoencapsulation promoted the synthesis and deposition of organized collagen fibers, a low quantity of blood vessels, and a high quantity of active fibroblasts in the connective tissue, indicating an efficient re-epithelialization process. Collectively, these findings may support the hypothesis that the *Caryocar brasiliense* oil biological action was improved when associated with chitosan NC [76]. 

Chitosan exhibits higher surface area and charge density when formulated at the nanoscale, improving re-epithelialization and formation of granulation tissue by modulating keratinocyte proliferation and migration [76,80,126]. In view of this, a study presented insulin-loaded chitosan nanoparticles, which were prepared following the ionotropic gelation methodology. The hydrogel was formed using Sepigel^®^ and tested in a diabetic wound model. As expected, the zeta potential of nanoparticles was positive (~40 mV), contributing to a wound nanoparticle adhesion. Significant improvements in the wound healing process were observed after the nanoencapsulation of active substances (Table 2). 

Bairagi and co-workers prepared Carbopol^®^ 980 hydrogel containing ferulic acid-loaded PLGA nanoparticles and investigated its effectiveness in a diabetic wound model [69]. The nanoprecipitation method was employed to prepare the nanoparticles, which were incorporated into the Carbopol^®^ gel matrix. To evaluate the healing effect, the ferulic acid was orally and topically administered (suspensions) and (hydrogel) to different animal groups. Both oral and topical treatments showed faster healing in comparison to diabetic control groups. Remarkably, the group that received both oral and topical treatments showed the best pharmacological effect. The results showed a significant increase in wound closure and an increase in hydroxyproline content as compared to other groups, suggesting the important contribution of topical therapy for wound contraction and granulation tissue formation. 

Some studies addressed the potentialities of NS as a wound healing enhancer, which was demonstrated by green tea polyphenol incorporation into NS. To investigate the effectiveness in a diabetic wound model, the suspension was converted into a PVA-alginate hydrogel hybrid system [79]. The findings showed that hydrogel containing green tea polyphenol NS improved the mobility of epidermal cells in vitro (Figure 8) when compared to hydrogel prepared with the non-encapsulated compound.

The application of simvastatin has been studied for wound healing purposes [72,89]. Previous data indicated that the drug increases vascular endothelial growth factor synthesis and release at the wound site, improving the epithelialization, restoring the physiological skin epidermal barrier, stimulating keratinocyte migration, and, consequently, closing the wound [72]. Simvastatin had already been incorporated into SLN [89] and NS [72]. Three Carbopol^®^ hydrogel formulations were studied, which had the following composition: (1) simvastatin-loaded NS; (2) levofloxacin; and (3) a mixture of simvastatin-loaded NS and levofloxacin at a ratio of 1:1. Levofloxacin was selected given its broad-spectrum antimicrobial action, avoiding wound healing impairments by infection. Interestingly, the hydrogel containing the mixture of simvastatin-loaded NS and levofloxacin seems to elicit better healing effects, suggesting that a microbial infection could retard the wound contraction [72]. 

Epidermal lipoxygenase enzyme extracted from *Ambystoma mexicanum* (AmbLOXe) is known to accelerate the process of wound healing. Despite this, AmbLOXe is a polypeptide and easily undergoes inactivation, losing its biological actions. To overcome this restriction, Oveissi and co-workers presented a wound device prepared by the incorporation of AmbLOXe-loaded pectin nanoparticles into a hydrogel of alginate [73]. Swelling is a desired characteristic for hydrogels intending wound healing because it provides exudates absorption and maintains the moist environment at the wound site [73]. A remarkable wound-healing effect of alginate hydrogel containing AmbLOXe-loaded pectin nanoparticles and alginate hydrogel containing empty pectin nanoparticles in comparison to normal saline was observed. Furthermore, re-epithelialization and reduction of abnormal scarring incidence in full-thickness wounds were also observed for alginate hydrogel containing empty pectin nanoparticles.

#### 3.4.2. Polymeric Micelles

Micelles are self-assembling nano-scaled spherical structures usually composed of phospholipids and/or polymers in aqueous dispersion. This system can load drugs by entrapping them in the core of copolymer micelles, enhancing the intrinsic water solubility [93]. While the core of micelles carries hydrophobic drugs, the external hydrophilic groups of the polymer stabilize the structure in an aqueous medium [66].

Curcumin has already been discussed herein, associated with NE, LP, and SLN [82,85,91]. In another approach, Alibolandi and co-workers prepared a wound device composed of dextran hydrogels and incorporated curcumin-loaded polylactic acid–polyethylene glycol (PLA–PEG) nanomicelles into the semisolid. Following, curcumin was associated with amphotericin B and incorporated into polymeric micelles. Then, drug-loaded micelles were gelled using genipin to prepare a micelle–hydrogel composite and evaluated its potential healing capacity in an excisional wound model [74]. The micelle–hydrogel was composed of two oppositely charged polypeptide-based micelle systems, the positively charged poly (L-lysine)-b-poly(phenylalanine) (PLL-PPA) and negatively charged poly(glutamic acid)-b-poly(phenylalanine) (PGA-PPA). In the wound environment, the tissue pH is acid (~4.5), a feature that can be explored to modulate the drugs from delivery systems and potentially improve the healing process. PGA chains become relatively uncharged and acquire a helical conformation, which strained the core of the micelles, resulting in a faster release profile of curcumin. This is required for preventing infection at the wound site [133]. On the other hand, PLL–PPA association exists in a charged random-coil state. The micelles’ structure and drug release profile remain the same, releasing the drug at a slow rate, and aiding wound healing. Polymeric micelles-based hydrogels also enhanced the wound-healing effect of *Thymus vulgaris* essential oil and insulin, in excisional [93] and diabetic wound models [81], respectively. Due to the presence of polymers with different surface charges, micelles can modulate the drug release rate by forming chemical bonds between polymer chains and the drug, changing the polymeric conformational structure, or even both mechanisms. In addition, the system could be pH-responsive, where at certain pH values the chemical bonds can break, interfering with the cross-linking degree of the semisolid matrix, and modulating the drug release and mechanical properties of the hydrogel [81]. In an environment containing elevated glucose concentration, a high degree of glucose is bound to polymer phenylboronic acid through hydrogen bonds, causing the disruption of the insulin micelles structure and inducing drug release [81].

## 4. Conclusions

Hydrogels have sparked scientific and pharmaceutical interest in recent years given their potential to act as a physical barrier to protect wounds and to provide an ideal environment for tissue regeneration through scaffolding. The loading of healing enhancer substances into hydrogels contributes to accelerating wound restoration by avoiding infections, modulating inflammatory processes, and other mechanisms. However, most drugs or active compounds present low solubility, poor skin penetration/permeation, and instability issues. Thus, the association of soft nanoparticles and hydrogels intended for the preparation of wound-healing dressings is a promising alternative.

Depending on the features of the carrier, different properties can be provided to the final formulation, such as enhanced bioadhesiveness, biocompatibility, permeation, and physicochemical stability. Remarkably, lipid-based systems have drawn attention, as they contribute significantly to the healing process by maintaining moisture in the wound bed through the occlusive film-forming effect. On the other hand, soft polymers demonstrated an important role in modulating drug release and permeation/penetration in the skin layers.

Furthermore, the use of natural materials to prepare the nanostructures and the hydrogels seems to positively impact the biological process of wound healing. Additionally, the potential interactions between the gelling agent and the type/composition of nanostructures are highly relevant and rarely investigated subjects that must be considered for a proper understanding of nano-based hydrogel advantages. 

Finally, the application of soft nanoparticles could contribute to developing novel wound-healing devices with optimized properties. Soon, smart-nano-base formulations with environmental response properties can be prepared as devices for wound healing, enabling increased bioavailability and concomitant delivery of drugs at the wound site.

## Figures and Tables

**Figure 1 pharmaceutics-15-00874-f001:**
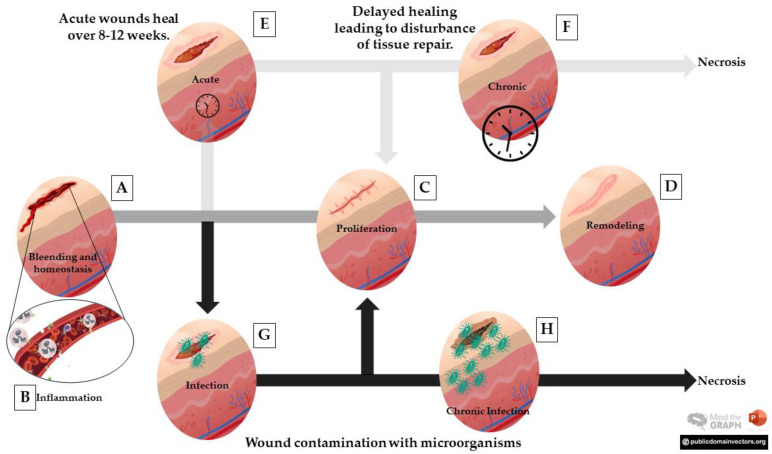
Flowchart of major wound healing outcomes. The wound healing process undergoes four stages: (**A**) Immediately after skin injury, the hemostasis initiates to stop bleeding; (**B**) Inflammation is mainly characterized by the presence of inflammatory cells in scar tissue and infection prevention; (**C**) The shift from the inflammatory phase to the proliferative phase occurs after fibroblasts’ migration to the wound, the accumulation of collagen and the formation of new endothelial structures inside the wound; (**D**) The last stage is the remodeling that consists of wound closure, organized collagen deposition, re-epithelialization, and scar tissue formation; (**E**) Acute injuries, such as surgical incisions, usually heal within days or weeks, and their extremities move closer together, thus decreasing the risk of infection; (**F**) Chronic wounds are defined as any interruption in the continuity of body tissue that hinders healing process and long-last permanence; (**G**,**H**) Infections can be hazardous to tissue, delaying the tissue restoration and increasing expenses, the duration of treatments, and the risk of complications. The healing process is influenced by various factors, including pressure, a dry environment, trauma, infection, and necrosis.

**Figure 2 pharmaceutics-15-00874-f002:**
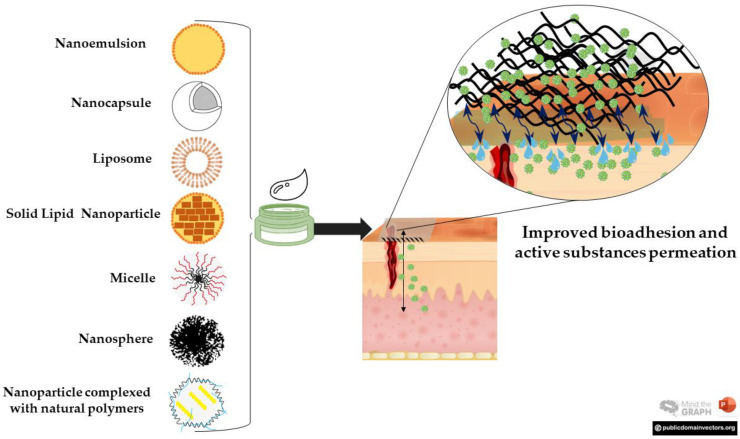
Nano-based hydrogels containing soft-particle-loaded active substances may improve the healing process. The major functions of wound dressings are to remove wound exudates, prevent the entry of harmful bacteria from the wound, and promote the establishment of the best milieu for natural healing. The association of nanocarriers can enhance the bioadhesiveness of the hydrogel and the permeation and penetration of actives at the wounded site, accelerating the healing process. The image was created using Mind the Graph platform (www.mindthegraph.com, accessed on 8 January 2022), Publicdomainvectors.org site (https://publicdomainvectors.org/, accessed on 8 January 2022), and Microsoft^®^ PowerPoint^®^ (Washington, DC, USA).

**Figure 3 pharmaceutics-15-00874-f003:**
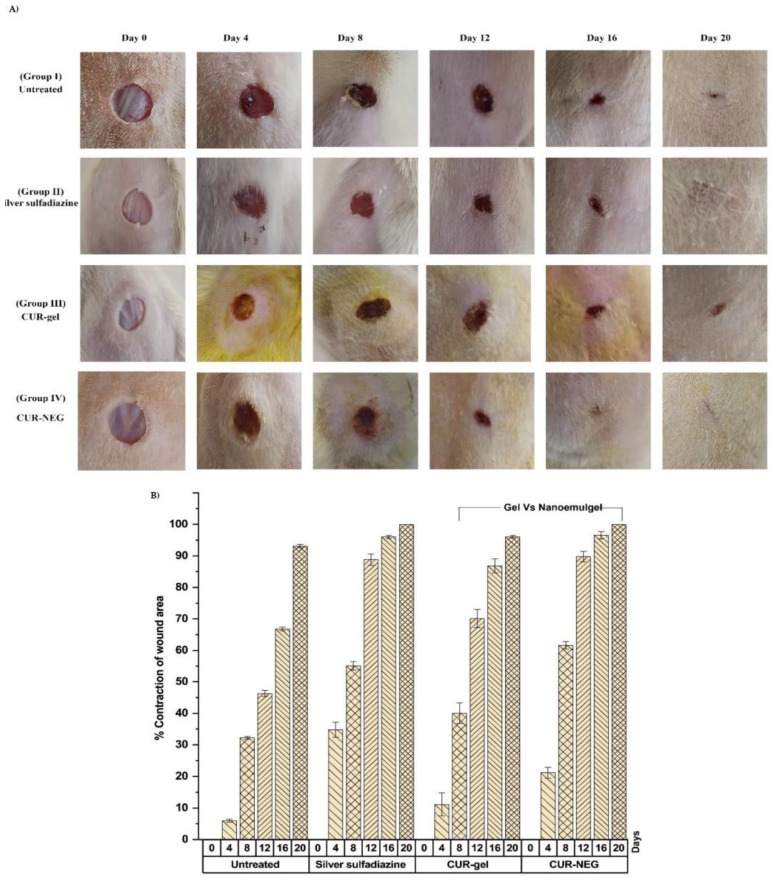
The healing effect was observed by % contraction of wound area over the treatments. (**A**) Representative images of wound healing in Wistar rats; (**B**) Percentage wound area contraction over the treatment. Groups identification: Group I received no treatment; Group II was treated with silver sulfadiazine semisolid formulation; Group III was treated with the hydrogel containing non-encapsulated curcumin (CUR-gel); Group IV was treated with a nano-based hydrogel of curcumin-loaded NE (CUR-NEG). Data are available at 10.3390/gels7040213 [85].

**Figure 4 pharmaceutics-15-00874-f004:**
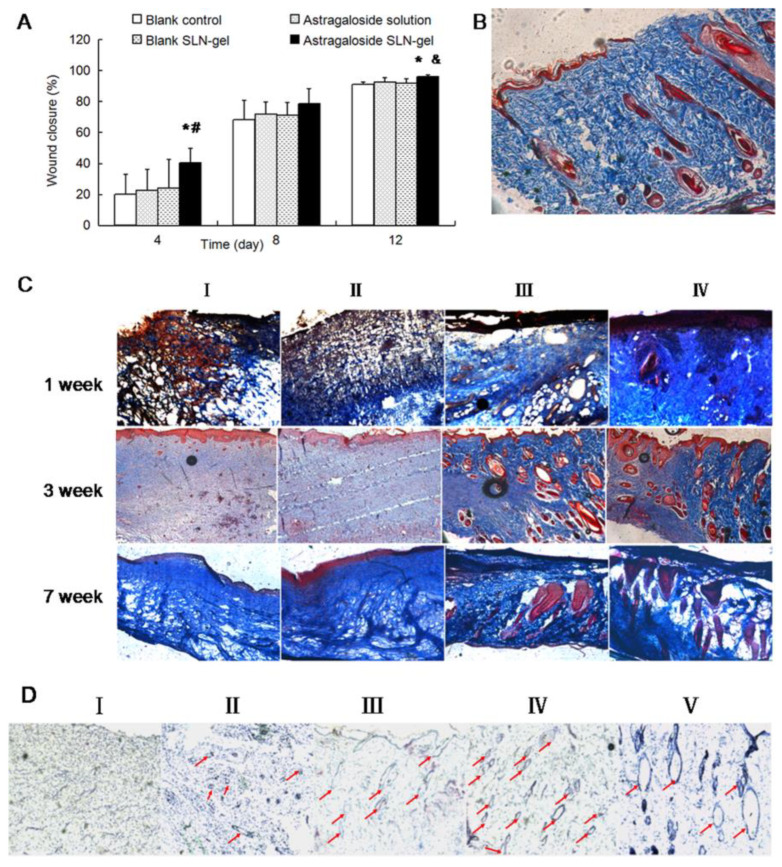
Effect of hydrogel containing Astragaloside IV-loaded SLN in wound healing and anti-scar potential after skin application. (**A**) Wound closure percentage (* *p* < 0.05, blank control; # *p* < 0.05, astragaloside IV solution; & *p* < 0.05, blank SLN-hydrogel). (**B**) Masson’s trichrome staining of normal skin. (**C**) Masson’s trichrome staining at 1-, 3-, and 7-weeks post-wounding: (I) blank control; (II) blank SLN-hydrogel; (III) astragaloside IV solution; and (IV) astragaloside IV SLN-hydrogel. (**D**) CD31 immunohistological staining at 3 weeks post-wounding: (I) blank control; (II) blank SLN-hydrogel; (III) astragaloside IV solution; (IV) astragaloside IV SLN-hydrogel; and (V) normal skin (red arrows indicate newly formed blood vessels). Figure reproduced with permission from Chen and co-workers [64].

**Figure 5 pharmaceutics-15-00874-f005:**
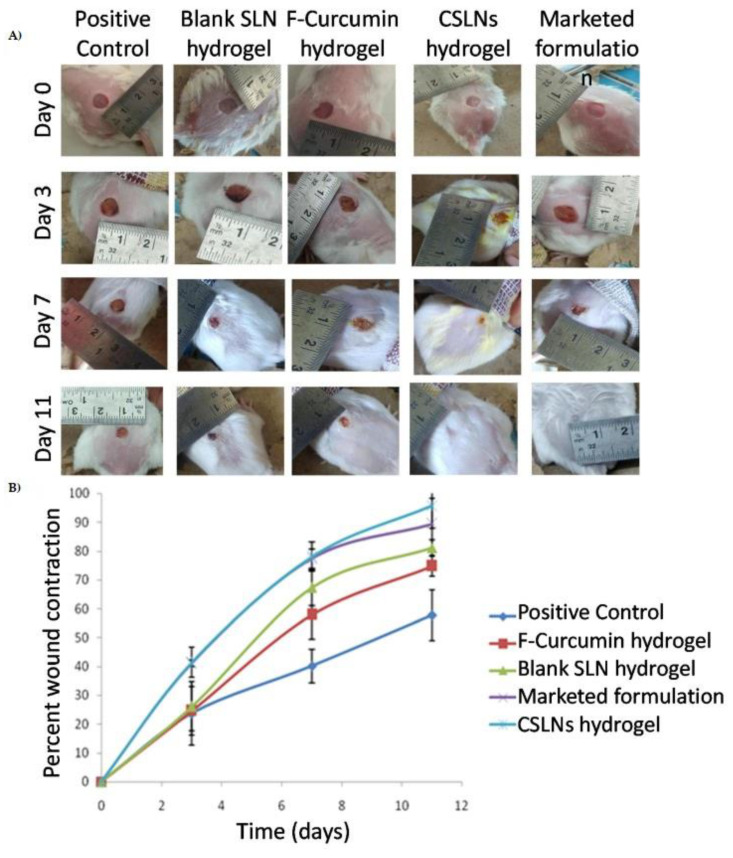
Effect of the treatments in the wound healing process. (**A**) Representative images of the wound healing process taken on day 0, day 3, day 7, and day 11 over the treatment; (**B**) Comparison of the percent wound contraction over the treatment. Data are available at 10.3390/antiox10050725 [82].

**Figure 6 pharmaceutics-15-00874-f006:**
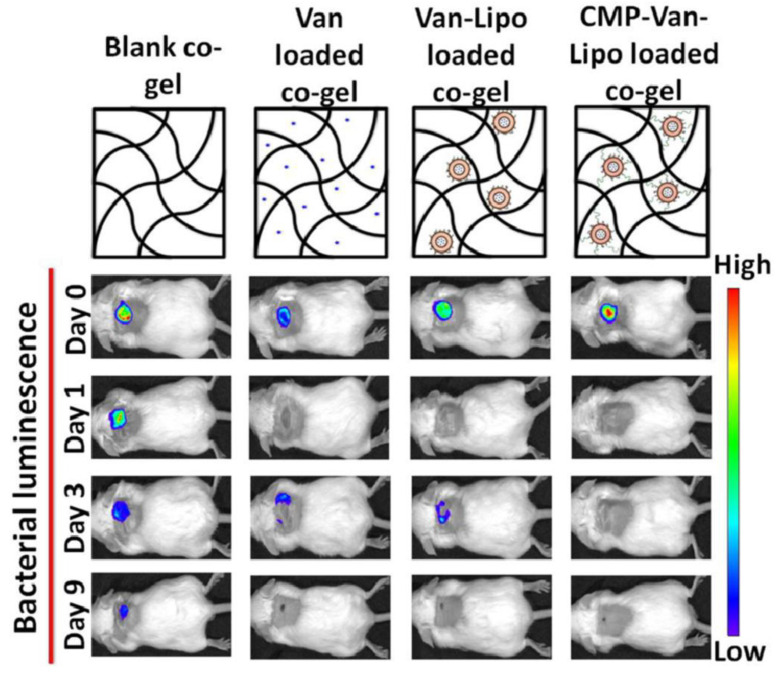
Lumina images for mouse punch biopsy wounds inoculated with luminescent *Staphylococcus aureus* followed by treatment with different groups: blank co-gels: hydrogel without Vancomycin; Van-loaded co-gels: hydrogel with non-encapsulated Vancomycin; Van-Lipo loaded co-gels: hydrogel with Vancomycin-loaded liposomes; CMP-Van-Lipo: hydrogel of collagen mimetic peptide conjugated vancomycin-loaded liposomes. Figure reproduced with permission from Thapa and co-workers [78].

**Figure 7 pharmaceutics-15-00874-f007:**
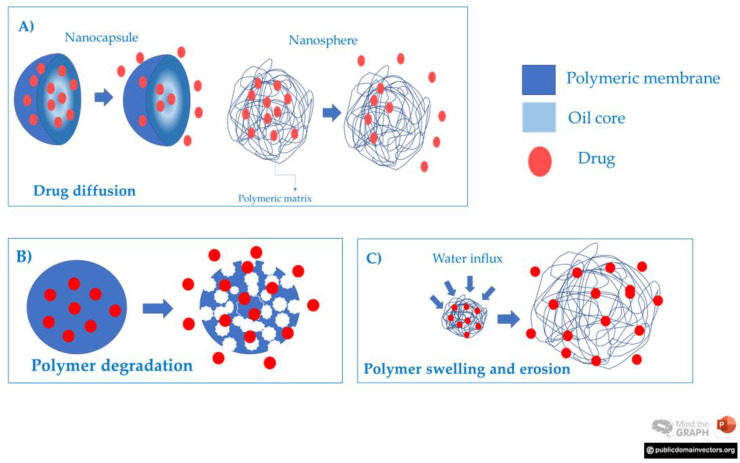
Mechanisms of controlled and sustained drug release from polymeric nanoparticles: (**A**) drug diffusion through the matrix, (**B**) polymer degradation, and (**C**) polymer swelling and erosion. The image was created using Mind the Graph platform (www.mindthegraph.com, accessed on 8 January 2022), Publicdomainvectors.org site (https://publicdomainvectors.org/, accessed on 8 January 2022), and Microsoft^®^ PowerPoint^®^ (USA).

**Figure 8 pharmaceutics-15-00874-f008:**
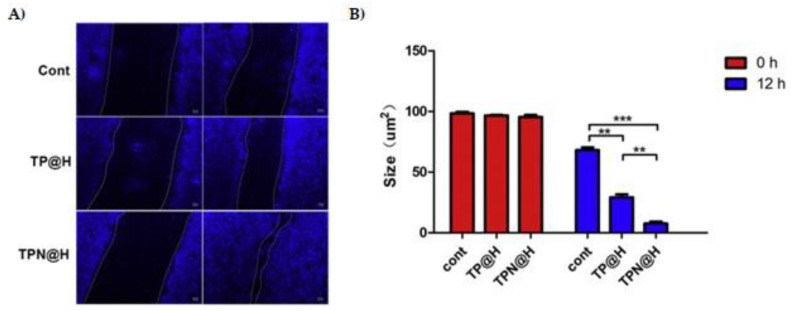
Cells-scratch wound healing in vitro assay. (**A**) Effects of TP@H (Hydrogel containing green tea polyphenol non-encapsulated) and TPN@H (Hydrogel containing green tea polyphenol-loaded nanospheres) on the migration of fibroblasts HFF-1 cells - Scale bar: 500 μm. (**B**) Quantitative analysis of the migration rates in (**A**). The blue fluorescence showed the nucleus, stained with DAPI The asterisks denote statistic differences with *p*-value *p* < 0.01 (**) or *p* < 0.001 (***). Figure reproduced with permission from Chen and co-workers [79].

**Table 1 pharmaceutics-15-00874-t001:** General aspects of the studies included in the review.

Reference	Active	System	Gelling Agent	Wound Protocol
Reimer et al., 2000 [61]	Povidone-Iodine	LP	Polyacrylic acid	Burn and chronic wound
Hurler et al., 2012 [62]	Mupirocin	LP	Chitosan	Burn Wound
Hurler et al., 2013 [63]	Mupirocin	LP	Chitosan	Burn Wound
Chen et al., 2013 [64]	Astragaloside IV	SLN	Carbomer 934	Wound scratch assay and Excisional Wound
Flores et al., 2015 [65]	Tea tree oil	NC and NE	Carbopol^®^ Ultrez 10 NF	Excisional Wound
Alibolandi et al., 2017 [66]	Curcumin	MC	Dextran	Excisional Wound
Jangde et al., 2018 [67]	Quercetin	LP	Carbopol^®^	Excisional Wound
Kakkar et al., 2018 [68]	Tetrahydrocurcumin	SLN	Carbopol^®^ 934	Excisional Wound
Bairagi et al., 2018 [69]	Ferulic acid	NS	Carbopol^®^ 980	Diabetic wound
Dawoud et al., 2019 [70]	Insulin	LP	Chitosan	Diabetic wound
Cardoso et al., 2019 [71]	Phenytoin	NC and NE	Chitosan	Excisional Wound
Aly et al., 2019 [72]	Simvastatin	PN	Carbopol^®^ 934	Excisional Wound
Oveissi et al., 2019 [73]	Ambystoma mexicanum epidermal lipoxygenase	PN	Alginate	Excisional Wound
Patel et al., 2019 [74]	Curcumin and amphotericin	MC	Genipin	Excisional Wound
Parisotto-Peterle et al., 2020 [75]	β-caryophyllene	NE	Hydroxyethylcellulose	Excisional Wound
Pires et al., 2020 [76]	*Caryocar brasiliense* oil	NC	HPMC	Excisional Wound
Back et al., 2020 [77]	Soybean isoflavone	NE	Carbopol^®^ Ultrez-21	Excisional Wound
Thapa et al., 2020 [78]	Vancomycin	LP	Collagen	Infected wound
Chen et al., 2020 [79]	Green tea polyphenol	NS	PVA/Alginate	Diabetic Wound
Ribeiro et al., 2020 [80]	Insulin	PN	Sepigel	Diabetic Wound
Zhu et al., 2020 [81]	Insulin	MC	Hyaluronic acid and chitosan	Diabetic Wound
Sandhu et al., 2021 [82]	Curcumin extract	SLN	Carbopol^®^ 934P	Burn Wound
Tahpa et al., 2021 [83]	Ethylenediaminetetraacetic	LP	Pluronic F127 and bacteriocin Garvicin KS (GarKS)	Infected wound
Algahtani et al., 2021 [84]	Thymoquinone	NE	Carbopol^®^ 940	Excisional Wound
Algahtani et al., 2021 [85]	Curcumin	NE	Carbopol^®^ 940	Excisional Wound
Ansari et al., 2022 [86]	Crisaborole	NE	Chitosan	Excisional Wound
Rehman et al., 2022 [87]	Eucalyptol	NE	Carbopol^®^ 940	Excisional Wound
Lee and Lin, 2022 [88]	Perfluorocarbon bromide and epidermal growth factor	NE and NP	Chitosan and PVA	Diabetic wound
Gupta et al., 2022 [89]	Simvastatin	SLN	Carbopol^®^ 934	Excisional Wound
Reczyńska-Kolman et al., 2022 [90]	Peptide—nisin	SLN	Gellan gum and Sodium Alginate	Cell migration and wound healing in vitro
Peralta et al., 2022 [7]	Miltefosine	LP	Carbomer 934-P NF	Cutaneous Leishmaniosis
Cardoso-Daodu et al., 2022 [91]	Curcumim	LP	Carbopol^®^ Ultrez	Excisional Wound
El-ezz et al., 2022 [92]	Copper (II) Schiff Base Quinoline	SLN	Carbopol^®^ 940	Excisional Wound
Alsakhawy et al., 2022 [93]	Thymus vulgaris essential oil	MC	Gelatin	Excisional Wound

Note: NE—nanoemulsions; NC—nanocapsules; SLN—solid lipid nanoparticles; LP—liposomes; MC—micelles; NS—nanospheres; PN—polymeric nanoparticles; HPMC—hydroxypropylmethylcellulose.

**Table 2 pharmaceutics-15-00874-t002:** Main findings of the studies and advantages conferred by nanosystems.

Reference	Main Findings	Formulations’ Hallmark Features
Reimer et al., 2000 [61]	A hydrogel of polyacrylic acid containing povidone-iodine-loaded phosphatidylcholine LP for achieving both antiseptic and moist treatment was prepared. The nanocarriers presented a multilamellar structure and particle size between 0.2 and 4.5 µm. An improved interaction with microorganisms (*Candida albicans* and *Staphylococcus aureus*) was achieved, which enhanced the antimicrobial action. The nano-based semisolid spreads easily and had significantly better tolerability in comparison with the conventional povidone-iodine ointment	Rate of neo-epithelization: NH > 90% × CP < 80%
Hurler et al., 2012; 2013 [62,63]	The liposomal suspensions showed an entrapment efficiency of mupirocin of 62.4% (±8.8%). The formulation was stable at different storage conditions (cold and accelerated studies) and presented a prolonged and controlled-release of mupirocin from the nano-based hydrogel. A satisfactory antimicrobial potential was obtained for chitosan hydrogel containing mupirocin-loaded phosphatidylcholine LP and an antibiofilm effect. The formulation presented promising effects in a mouse model of burn	Retention of formulation on the skin surface: NH~50% × CP < 20% Wound size after 20 days: NH~4 nm × H 4–6 nm × CP~6 nm
Chen et al., 2013 [64]	Astragaloside IV was incorporated into SLN, using glycerol tristearate as a lipid component, and further thickened in a Carbopol^®^ 934 hydrogel. Astragaloside IV-based SLNs enhanced the proliferation and migration of keratinocytes, indicating its capability to accelerate wound re-epithelialization	Wound scratch assay—wound closure (%): SLN > drug solution > C In vivo model—wound closure (%): NH > BH > drug solution > C
Flores et al., 2015 [65]	*Melaleuca alternifolia* essential oil was incorporated into NC and NE to prepared Carbopol^®^ Ultrez hydrogel for investigating its potential application in wound healing. NC-based semisolid was more promising in comparison to the NE. The formulation reduced the wound area and accelerated re-epithelization and angiogenesis to a greater extent than NE-based hydrogel	Spreadability factor (mm^2^/g): NCH 4.53 × NEH 4.78 × FH 2.85 × BH~3.70 Wound area reduction (%): NCH >50% × NEH < 40% × FH~40% × BH~25%
Alibolandi et al., 2017 [66]	Curcumin-loaded polylactic acid–polyethylene glycol (PLA–PEG) nanomicelles were prepared and converted into a dextran hydrogel. The water absorption property of dextran hydrogel lasted until eight days after its preparation. Curcumin was slower released when nanomicelles were incorporated into hydrogel than the suspension form. Curcumin nanomicelle/dextran hydrogel presented a sustained drug release from a hydrated device, reducing inflammatory responses, promoting fibroblast proliferation and collagen production, and improving angiogenesis in wound healing	Collagen deposition: NH > BH > C Wound area (cm^2^): NH < BH < C
Jangde et al., 2018 [67]	The nano-based formulations had a water vapor transmission rate suitable for assisting in maintaining the humectants’ properties of the skin. The quercetin-loaded LP hydrogel was found to be stable under both normal and accelerated conditions, hemocompatible, and presented an adequate swelling ratio to the management of wounds	Wound contraction after 12 days: NH 98.93% × CP 92.97% × control 76.83% Epithelization period (days): NH 16 × CP 18 × control 26
Kakkar et al., 2018 [68]	The tetrahydrocurcumin-loaded SLN (Compritol^®^ 888 ATO and Phospholipon 90G as a lipid matrix) hydrogel formed an intact lipid film on the filter paper surface, suggesting a possible occlusion effect on the formulation, which could be attributed to the solid nature of the lipid components. Otherwise, vehicle Carbopol^®^ 934 hydrogels had the lowest occlusion factor due to the absence of any lipids in the formulation	Occlusion factor after 48 h (%): NH~83 × BH~82 × H~23 × CP~2
Bairagi et al., 2018 [69]	Ferulic acid-loaded polymeric nanoparticles dispersion (oral administration) and ferulic acid-loaded polymeric nanoparticles based hydrogel (topical administration) treated wounds were found to epithelize faster as compared with the diabetic wound control group. Both oral and topical treatments showed faster healing in comparison to diabetic control groups. The group that received both oral and topical treatments showed the best pharmacological effect. Nanoparticles presented a biphasic drug release profile, consisting of an initial burst release followed by sustained release up to 48 h	Wound area (cm^2^): NH < CP < FH
Dawoud et al., 2019 [70]	The LP (egg phosphatidylcholine and cholesterol) presented particle size around 250 nm, negative zeta potential values, and 87.379% entrapment efficiency of insulin. The nano-based hydrogel presented high physicochemical stability, suitable viscosity, and sustained the insulin release for up to 24 h. In a diabetic wound healing model, an improvement of 16-fold in the healing rate in comparison to the control group was observed, suggesting a highly stable and therapeutically promising drug delivery system for diabetic wound management	In vitro drug release: FH 6 h × NH 24 h Healing rate: NH 36.67 ± 12.179 mm^2^/day × H 36.67 ± 12.179 mm^2^/day
Cardoso et al., 2019 [71]	Phenytoin was incorporated into an NC (poly-epsilon-caprolactone and grape seed oil) or NE system (grape seed oil). A superior content of phenytoin reached the dermis layer from the liquid form of NE in comparison to NC. As expected, the drug penetration increased using injured skin, reaching the epidermis and dermis layers	Consistency index (Pas^n^): NCH 24.23 × NEH 24.53 × FH 55.25 BH~17 Work of adhesion (mN.mm): NCH 9.5 × NEH 10.3 × FH 1.1 Wound area contraction (%) after day 6: NCH~0.6 × NEH > 0.6 × FH < 0.6
Aly et al., 2019 [72]	The simvastatin-loaded NS (PEG 4000) Carbopol^®^ 940 hydrogel showed significant results in the healing process, presenting complete epithelialization, minimal inflammatory cell infiltration, mature collagen fiber formation, and more activated hair follicle growth after 11 days of the protocol	Wound contraction (%): NH~90 × C~80
Oveissi et al., 2019 [73]	The alginate hydrogel containing epidermal lipoxygenase enzyme-loaded pectin nanoparticles had a swelling ratio higher than empty alginate hydrogel. Nano-based formulation demonstrated better mechanical properties than empty hydrogel. A sustained delivery of drug from nano-based hydrogel was achieved in comparison to drug solution and hydrogel containing non-encapsulated pectin	The score of wound re-epithelialization: NH 1.66 × BH 0.66 × C 0.33
Patel et al., 2019 [74]	The study applied a dual-drug release system composed of polylactic acid–polyethylene glycol to deliver an anti-bacterial drug (amphotericin B) during the early stages of healing, followed by a slow release rate of the healing enhancer drug (curcumin). Better healing was observed in rats treated with the drug-loaded hydrogels than in the blank and control groups. Wounds showed up to 80% closure in the treated group, with high collagen deposition. Re-epithelialization and granulation were also better in the treated group than in the non-treated control and blank groups	Wound closure (%): NH~87 × BH~33 × C19
Parisotto-Peterle et al., 2020 [75]	The hydroxyethylcellulose hydrogels containing β-caryophyllene-loaded NE presented 892.40 ± 81.53 mN and 444.37 ± 78.82 mN of maximum force for detachment in intact skin and injured skin (stratum corneum removal), respectively. In vivo healing study revealed that treatment with the NE-based hydrogel accelerated wound closure (with reduced surface area and wound contraction) and that the effect may be related to a decrease in the acute inflammatory process as evidenced by a decrease in IL-1, TNF-α, and MPO markers	Drug permeation: *Stratum corneum*: drug solution > NE > NH Epidermis: NE = NH > drug solution Dermis: NE > NH > drug solution
Pires et al., 2020 [76]	The obtained hydrogels were homogeneous after associating the nanosystems (poly-episolon-caprolactone lipid core NC), presenting suitable topical applicability and spreadability, with a viscosity ranging from 8130 and 11,740 cp. The lesions’ average area of the nano-based hydrogel was 0.043 cm^2^, which was 400-fold lower than the other groups. Notably, no lesions were observed after 14 days of treatment	Lesion area (cm^2^): NH 0.043 × BH 0.454 × FH 0.634 × CP 0.131
Back et al., 2020 [77]	Soybean isoflavone aglycones(IAF)-loaded NE dispersed in acrylic-acid hydrogels have been investigated as potential wound healing compounds for cutaneous application. The findings demonstrated the potential of nano-based hydrogels to promote wound healing by increasing angiogenesis, reducing lipid oxidation, and inflammation.	Drug permeation in intact skin: *Stratum corneum*: NH > NE > drug solution Epidermis: NH > NE > drug solution Dermis: NH > NE > drug solution Drug permeation in skin without stratum corneum: Epidermis: NE > NH > drug solution Dermis: NE > NH > drug solution
Thapa et al., 2020 [78]	The LP was composed of 1,2-dipalmitoyl-sn-glycero-3-phosphocholine (DPPC):Cholesterol:1,2-distearoyl-snglycero-3-phosphoethanolamine-N-[methoxy(polyethylene glycol)-2000-maleimide] at a molar ratio of 73:24:3 and had 150 nm of diameter with a low PDI (0.05) and negative zeta potential (−3.0 mV). The LP-based semisolid formulation exhibited desirable viscosity and rheological properties intending topical applications along with the controlled release of the peptide (up to nine days). Additionally, potent in vitro antibacterial and anti-biofilm effects devoid of fibroblast toxicity were assigned to the nano-based hydrogel gel. The mouse model of infected wounds demonstrated potent antibacterial effects of nano-based hydrogel	Antibacterial effects in wound site: NH up to 9 days × FH < 2 days
Chen et al., 2020 [79]	The PVA-alginate hydrogel containing green tea polyphenol NS enhanced the mobility of epidermal cells in vitro. In the diabetic wound model, hydrogel containing green tea polyphenol-loaded NS provided wound closure faster than control and vehicle hydrogel groups. Molecular investigations indicated that nano-based hydrogel containing green tea polyphenols may reduce inflammation and enhance diabetic wound healing by activating the PI3K/AKT pathway	Wound size (µm^2^)—in vitro: NH < FH < C
Ribeiro et al., 2020 [80]	Insulin-loaded chitosan nanoparticles were prepared by the ionotropic gelation method. Healing evaluations showed that polymorph nuclear infiltrate was more significant for blank- and insulin-chitosan nanoparticles, demonstrating that chitosan exerted chemotaxis of these cells, stimulating their migration to the wound	Degree of wound: CP > NH > FH > BH
Zhu et al., 2020 [81]	The hydrogels containing insulin-loaded micelles and epidermal growth factor showed an excellent wound healing performance for the promotion of fibroblast proliferation and tissue internal structure integrity, as well as the deposition of collagen and myofibrils	Wound area (%): NH < BH < Gauze
Sandhu et al., 2021 [82]	The curcumin-loaded SLN (Compritol^®^ 888 ATO and Phospholipon 90G) association with hydrogel extended the phytochemical release for up to 120 h. In the excisional model, on day 11, hydrogel resulted in complete wound closure, which was superior to the positive control, blank SLN hydrogel, and non-encapsulated curcumin hydrogel	Wound contraction (%): NH~96 × CP~58 × BH~81 × FH~75
Thapa et al., 2021 [83]	The phosphatidylcholine LP containing antibacterial peptides presented homogenous size distributions with an average particle size of ≈180 nm and PDI below 0.2. The Pluronic F127 hydrogel exhibited suitable viscosity and rheological properties along with controlled release behavior (up to nine days) for effective peptide delivery following topical application. Potent in vitro antibacterial and anti-biofilm effects of bacteriocin Garvicin KS (GarKS) gel were evident against the Gram-positive bacterium *Staphylococcus aureus*. The in vivo treatment of methicillin-resistant *S. aureus* infected mouse wounds suggested potent antibacterial effects of the GarKS gel following multiple applications of once-a-day application for three consecutive days	Anti-biofilm effects: FH~75% × NH~82% × BH~18%
Algahtani et al., 2021 [84]	The nanoemulgel system of thymoquinone-loaded black seed oil NE exhibited significant enhancement in skin penetrability and deposition characteristics after topical administration compared to the conventional hydrogel system. The developed nanoemulgel system of thymoquinone exhibited quicker and early healing in wounded Wistar rats compared to the conventional hydrogel of thymoquinone	Complete epithelialization (days): C~16 × CP~11 × FH~14 × NH~10
Algahtani et al., 2021 [85]	The developed curcumin nanoemulgel (Labrafac PG and PEG 400) exhibited thixotropic rheological behavior and a significant increase in skin penetrability characteristics compared to curcumin dispersed in a conventional hydrogel system	Cumulative Amount of Drug Permeated (µg/cm^2^): NH 773.82 × FH 156.90 Drug Deposited in Skin (µg/cm^2^): NH 1161.54 × FH 179.47 Permeability (K Coefficient × 10^−3^): NH 5.49 × FH 0.876
Ansari et al., 2022 [86]	The chitosan gel containing crisaborole-loaded NE (Lauroglycol-90^®^) exhibited a flux of 0.211 mg/cm^2^/h, a drug release of 74.45 ± 5.4% in 24 h with a Korsmeyer-Peppas mechanism release behavior. The nano-based hydrogel presented promising wound healing and anti-inflammatory actions	Epithelialization period (days): C~28 × CP~20 × NH~21
Rehman et al., 2022 [87]	Carbopol 940 hydrogel containing eucalyptol-loaded NE (black seed oil) exhibited pH values between five and six, and acceptable homogeneity and spreadability	Wound contraction (%) after 15 days: NH 100.00 × CP 98.17 × C 70.84
Lee and Lin, 2022 [88]	Chitosan-based hydrogel associating perfluorocarbon (PFC)-loaded NE, epidermal growth factor (EGF)-loaded chitosan nanoparticles, and polyhexamethylene biguanide was prepared for treating diabetic wounds. The formulation could sustainably release the actives in an ion-rich environment to exert antibacterial effects and promote cell growth for wound repair	Degree of wound closure after 15 days: NE + NPH > NPH > NEH
Gupta et al., 2022 [89]	Significant loading of simvastatin (10% *w*/*w*) was achieved in spherical nanoparticle Carbopol 934 hydrogel (0.3 nm (nanoparticles) to 2 µm (gelled-matrix)) that exhibited good spreadability and mechanical properties and slow release up to 72 h. Complete healing of excision wounds observed in rats within 11 days was 10 times better than the commercial povidone-iodine product	Wound closure (%) after day 11: NH > 90 × FH < 50 × CP < 30
Reczyńska-Kolman et al., 2022 [90]	Encapsulation of nisin into stearic acid-based SLN slowed its release from gellan gum-based hydrogels for up to 24 h. The most effective antimicrobial activity against Gram-positive *Streptococcus pyogenes* was observed for the nanoformulation. All materials were cytocompatible with L929 fibroblasts and did not cause an observable delay in wound healing	Drug release after 24 h: NH~70 × FH~90%
Peralta et al., 2022 [7]	The Carbomer 934 hydrogel containing Miltefosine-loaded LP eliminated 99% of the parasites and cured the lesions with a complete re-epithelisation, no visible scar, and re-growth of hair. Fluid liposomes decreased the time to heal the lesion and the time needed to eliminate viable amastigotes from the lesion site	Lesion size (mm^2^) after 3 weeks: NH~0 × FH < 10 × C > 40 Parasitic inhibition (%): NH 99.66 × FH 99.80
Cardoso-Daodu et al., 2022 [91]	The in vivo wound healing evaluation showed that curcumin-loaded LP (phosphatidylcholine) + lysine and collagen in Carbopol hydrogel had the highest percentage of wound contraction at 79.25% by day three post-surgical operation. The histomorphometric values show the highest percentage of collagen, lowest inflammatory rates, highest presence of microvessels, and re-epithelization rates at the wound site	Wound contraction: NH 79.25% day 3 × BH 23.70% day 7 × C < 60% day 3 Microvessels in granulation tissue (vessels/mm^2^): NH~4 × BH~2 × C~1
El-ezz et al., 2022 [92]	Carbopol 940 hydrogel loaded with Cu (II) Schiff base 8-hydroxy quinoline complex (CuSQ) SLN (soy lecithin and cholesterol) showed good homogeneity and stability, a pH of 6.4, and had no cytotoxicity on the human skin fibroblast cell line. The nano-based formulation showed significantly faster healing rates compared to standard and control rats	Wound healing (%): NH > BH > CP = C
Alsakhawy et al., 2022 [93]	The release profile of thyme essential oil-loaded nanocomposite hydrogel revealed a sustained release pattern compared to thyme essential oil-loaded micelles and free oil. The thyme essential oil-loaded nanomicelles exhibited a significantly higher antibacterial effect than the non-encapsulated compound. Furthermore, the nano-based hydrogel significantly promoted wound contraction, reduced interleukin-6, and increased transforming growth factor-β1 and vascular endothelial growth factor levels, versus control or blank hydrogel group	Inflammation: NH < BH < C Formation of blood vessels: NH > BH > C

Note: NE—nanoemulsions; NC—nanocapsules; SLN—solid lipid nanoparticles; LP—liposomes; MC—micelles; NS—nanospheres; PN—polymeric nanoparticles; HPMC—hydroxypropylmethylcellulose; H—hydrogel without nanocarrier; NH—nano-based hydrogel; FH: hydrogel containing non-encapsulated (free) drug; BH—hydrogel containing blank nanostructures; CP—commercial product; C—Control.

**Table 3 pharmaceutics-15-00874-t003:** Wound contraction percentage over the treatment schedule with nano-based hydrogel, commercial formulation, and vehicle.

Days	Control	Nano-Based Hydrogel	Commercial Product
3rd day	05.106% ± 0.110	25.633% ± 0.549	15.146% ± 0.254
5th day	20.050% ± 0.055	40.483% ± 0.422	30.236% ± 0.409
7th day	25.433% ± 0.388	55.536% ± 0.474	46.590% ± 0.510
9th day	40.233% ± 0.245	60.583% ± 0.514	60.326% ± 0.473
11th day	55.313% ± 0.419	80.443% ± 0.403	78.026% ± 0.380
13th day	60.420% ± 0.435	90.430% ± 0.603	89.253% ± 1.090
15th day	70.846% ± 0.830	100.000% ± 0.015	98.170% ± 0.749

Control: formulation without Eu; Nano-based hydrogel: hydrogel containing Eucalyptol-loaded nanoemulsion; commercial product: Quench Cream ^®^, containing silver sulphadiazine. The data are presented as mean ± SD and analyzed using one-way ANOVA. *p* < 0.02 and *p* < 0.04 refer to the statistical significance of both groups from the control. Data are available at 10.3390/pharmaceutics14091971 [87].

## Data Availability

Data is contained within the article.

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
