# Peer review of "Status and Future Scope of Soft Nanoparticles-Based Hydrogel in Wound Healing"

_pharmaceutics, 2023, doi:10.3390/pharmaceutics15030874_

Round 1
Reviewer 1 Report
Title: Status and future scope of soft nanoparticles-based hydrogel in wound healing
Comments: The review is very extensively conducted which covers the basis of the types of wounds to the usefulness of the nanoparticles in the in-vivo environment.
1. The study objectives are defined. The clinical relevance and rationale of this study are described.
2. In the literature search, a comprehensive literature search is conducted and presented in a funnel-type fashion which is easy to read and understand
a. The images need to be worked on such as figure 2; where the alphabets can be more clear.
b. Adaptation of the images needs to be present
Author Response
"Please see the attachment."

Reviewer 2 Report
This is a very informative and well-organized review. The choice of references is fine, and the representation of the result has an added value for the navigation in the rapidly growing and multidisciplinary field of wound healing. The review is interesting both for chemical engineers and biomedical researchers.
The only week point is the illustration. Figures 1 and 2 are not detailed enough, and it is not a good choice to put most of the information in the caption instead of making the cartoons more detailed. Furthermore, the last chapters are not illustrated at all. Five figures in this long review are just not enough to help the reader. This should be addressed reviewing the manuscript, which is expected to significantly improve the appeal of the work towards the targeted audience.
Author Response
"Please see the attachment."

Reviewer 3 Report
This is a wonderful comprehensive review of the use of nanoparticle based hydrogels in wound healing. My primary concern is organization One would hope for a clearer exposition of the critical differences between nanoparticle based and non nanoparticle based hydrogels. The authors have presented an exhaustive narrative, but one would like to see a table comparing ostensible performance of the various approaches with nanoparticle as compared to non-nanoparticle based hydrogels.
In doing so, the authors must acknowledge the extreme difficulty of demonstrating superiority of wound healing modalities compared to active control agents. Perhaps a column in their table could state the evidence for concluding incremental benefit of the nanoparticle system for each agent.
Finally, one must deal with the issue of negative control. Is it conceivable that then nanoparticle material itself provides benefit without any incorporated substance?
SPECIFIC COMMENTS
to feasible the application TO OPTIMIZE
Currently, a wide variety of dressings suitable for 52 managing both acute and chronic wounds are available,
PLEASE PROVIDE A TABLE OF ALL THE MANY WOUND DRESSINGS WITH NANOPARTICLE FORMULATION AND WITHOUT NANOPARTICLE FORMULATION
where hydrogels are particularly 53 highlighted for their potential [13].
IN THE PRESENT MANUSCRIPT. WE FOCUS ATTENTION ON THE POTENTIAL ADVANTAGES OF HYDROGELS
allowing clinical monitor the
ALLOWING CLINICAL MONITORNG OF
extremities are not ap-114 proach each other, i
DO NOT APPROACH EACH OTHER
There are some lesions caused by infective diseases, such as Leishmaniasis and its 134 cutaneous form [35].
SUCH AS, FOR EXAMPLE, LEISHMANIASIS, QUITE FREQUENTLY ENCOUNTERED IN SOUTH AMERICA
pathogens. Following, monocytes achieve the injured site and are activated into 195 macrophages,
FOLLOWED BY MONOCYTES
Remarkably, macro-197 phages are fundamental for the transition to the
DELETE “REMARKABLY”
adverse effects trig-261 gered by drug toxicity [17
PLEASE ELABORATE
ease carriers 271 delivery through the skin,
EASE OF DELIVERY
because allows the maintenance
BY ALLOWING THE MAINTENANCE
Regarding polymeric particles, the presence of polymers confers 349 some advantages for hydrogels, such as prolonged drug release, enhanced stability, im-350 proved mechanical strength, and increased
CAN YOU GIVE EXAMPLES OF A POLYMERIC AND A OF A NON-POLYMERIC NANOPRTICLE. TABLE I IS CONFUSING BECAUSE IT LABELS ‘HYDROGEL POLYMER” FOR ALL ENTRIES
Differently from conventional emulsion, NE is referred to as dispersed systems with 375 ≤ 100 nm droplets, high thermodynamics, and stability properties [87,88].
WHY IS THAT DIFFERENT?
Natural products present several medicinal potentials for wound 379 management, e.g., antimicrobial, anti-inflammatory, analgesic, and antioxidant proper-380 ties. Despite this, they are still poorly used for treating skin lesions due to the low perme-381 ation through skin layers, inadequate molecular size, and physicochemical instability [90
CAN YOU CLARIFY WHAT YOU MEAN BY “POORLY USED”?
Zeta potential values of nanoparticles
WHAT IS THE ZETA POTENTIAL?
Author Response
"Please see the attachment."

Reviewer 4 Report
The manuscript entitled Status and future scope of soft nanoparticles-based hydrogel in wound healing is a very thorough review covering the studies of hydrogels containing soft-particles loaded with active substances for wound healing. In my opinion, the Table that summarizes each of the evaluated studies will be extremely useful for the research community. Minor: There is a large discrepancy in the English language between the Abstract and the rest of the manuscript. This should be corrected, for example:- line 16, "easiness of application" should be "ease of application"
- lines 24-25, the sentence should be rewritten
- line 26, "drugs-based hydrogels" should be "drug-based hydrogels"
- lines 27-28, "we expect to contribute to improving the pharmacological management of wounds" should be rewritten
Line 1099, change "agent gelling" to "the gelling agent"
Line 1102, replace "then" with "finally"
Line 1103, replace "in a near future" with "in the near future"
Reviewer 5 Report
The manuscript authored by Luana Mota Ferreira and co-workers reports a review article about soft nanoparticle-based hydrogels, including nanoemulsions, solid lipid nanoparticles, liposomes, and polymeric systems, for wound healing applications. Overall, I consider the review could be of interest to the readers of Pharmaceutics, but the following major revisions need to be addressed by the authors to make the manuscript more attractive for the readers.
1) I suggest moving the section 2.3 to the beginning of the section 3.
2) I recommend erasing the paragraph (lines 296-306) on page 7 because these numerical data do not collect all published articles within this scope.
3) Ketoprofen and desketoprofen are widely employed as anti-inflammatory drugs for the treatment of inflammatory diseases, an important process that influences wound healing. In this regard, I suggest discussing this effect and including some works about polymer nanoparticles and hydrogels as carriers of desketoprofen and ketoprofen, DOI: 10.1021/acsami.1c22993; DOI: 10.2147/IJN.S140934; DOI: 10.1021/acsomega.2c07232; DOI: 10.1016/j.msec.2017.08.025.
4) I suggest deleting Table 2 because it is too dense and does not summarize the main findings. I consider these long main findings can be discussed in the text.
5) I also suggest organizing the information in sections 3.1, 3.2, 3.3, and 3.4 in sub-sections to make the text more easily readable and attractive.
6) The labeling of the figures needs to be homogenized to use always the same format, small (a) or capital (A) letters. Authors should choose one format.
7) The references section needs to be double-checked. Some information is duplicated in some references, article numbers or article pages are missing, and other references are incomplete. Please revise references 5, 16, 17, 23, 24, 29, 34, 38, 39, 43, 49, 59, 60, 61, 64, 65, 66, 67, 70, 72, 74, 98, 101, 110, 111, 114, 117, 118.
Round 2
Reviewer 3 Report
This is a significant contribution. My primary concern is organization. The addition of the last column to Table 1 is an improvement, but much more narrative explanation is required. In particular, there is a need for comparison of the gelling agents. How is chitosan different from Carbopol from hydroxycellulose, from alginate from....? That discussion is missing. There is a great deal of narrative in this article, but the narrative should focus on the Table, rather than being extemporaneous.
Author Response
This is a significant contribution. My primary concern is organization. The addition of the last column to Table 1 is an improvement, but much more narrative explanation is required. In particular, there is a need for comparison of the gelling agents. How is chitosan different from Carbopol from hydroxycellulose, from alginate from....? That discussion is missing. There is a great deal of narrative in this article, but the narrative should focus on the Table, rather than being extemporaneous.
Answer: We would like to thank the comments. In the revised version, we focus on improving objectivity and better exploring the selected articles and their findings. To this end, a novel table (Table 2) was elaborated in order to enhance comprehension. The main findings were highlighted in the second column, summarizing the results of each study, while appropriate discussions were maintained as narrative in the topics. Formulations' hallmark features were presented in the third column, reporting the performance of nano-based hydrogels, as previously requested by the reviewer in the first round of revision. After these modifications, we hope that the document meets the expected standard regarding clarity, organization, and comprehension as demanded by the reviewer.
Even considering that comparison of the gelling agents is out of the scope of our review, their potential contributions of them were discussed throughout the manuscript (highlighted in green) when appropriated for better comprehension of the hydrogels’ performance. In addition, most of the materials used for preparing the hydrogels approached herein are widely investigated and exhaustively described in the scientific literature. In view of this, our focus was the influence of soft nanoparticles on general hydrogel properties as a strategic novel approach to managing wounds. Despite this, we thank the reviewer for the pertinent appointment, and we will consider studying the potential interactions between agent gelling and the type/composition of nanostructures. In our conclusion section we included this topic as a perspective in future investigations.
Lastly, we would like to reinforce that all document was detailly revised and polished concerning English language and syntax, and typographic issues to enhance its readability. The alterations were highlighted in red.

Reviewer 5 Report
The authors have addressed the suggested revisions and the manuscript can be accepted for publication in its current form.
Round 3
Reviewer 3 Report
the authors are declining to discuss the comparative properties of the various gelling agents claiming "that is out of scope". This reviewer strongly disagrees. A relatively brief summary of the the comparative properties would markedly improve this manuscript.
Author Response
Dear reviewer,
We respectfully receive your concern about the lack of additional discussions regarding gelling agents. In the previous rounds of revision, we tried to improve this topic by including brief discussions when pertinent throughout the paper. In the revised version, the main features of gelling agents in the context of each study were mentioned. However, we do not agree with prioritizing this subject given its complexity, which would considerably modify the purpose of our study. We recognize the importance of exploring the gelling agents and their contribution to the final properties of formulations, but it is out of our goals. Moreover, it is relevant to mention that the modifications requested were detailed approached in a recent publication of another MDPI journal (DOI: 10.3390/gels8030174), reinforcing the importance of maintaining the focus on our main objectives, which are novel and highly relevant for the scientific community.
Lastly, we greatly appreciate your suggestions during the revision process, which certainly contributed to improving the quality of our manuscript and we deeply apologize if we do not properly attend to your request.
Best regards